


# Probabilistic seismic risk assessment for cities: Counterfactual analysis in a Chilean case study

Rosita Jünemann[1,2], Alejandro Urrutia[1], Monserrat Damian[1], Oscar Ortiz[1], Felipe Zurita[1,3], and Jorge G.F. Crempien[1,2]

[1]Research Center for Integrated Disaster Risk Management (CIGIDEN), Santiago, Chile
[2]Escuela de Ingeniería, Pontificia Universidad Católica de Chile, Santiago, Chile
[3]Instituto de Economía, Pontificia Universidad Católica de Chile, Santiago, Chile

**Correspondence:** Rosita Jünemann (rjunemann@ing.puc.cl)

**Abstract.** We develop a comprehensive probabilistic seismic risk assessment model and apply it to the coastal city of San Antonio in Chile. We use this model to analyze the implications of various counterfactuals of the exposed building stock, in terms of different risk metrics. We begin by generating a synthetic earthquake catalog via Monte Carlo simulations. We then simulate spatial seismic intensities by using ground motion models, considering correlation parameters specifically estimated for the Chilean subduction zone, to evaluate ground motion intensity measures at each site. Additionally, we develop a high-resolution exposure model that characterizes the exposed building stock and population distribution at the census block level. We use the proposed methodology to obtain risk curves for the current building stock, as well as for four counterfactuals involving changes in building materials and/or design levels. Thus, we quantitatively identify the most effective alternatives for mitigation plans. These alternatives consider not only physical damage but also economic losses and casualties, showing the method's potential for its use as a valuable public policy planning tool for decision-makers. Specifically, we find that changing either the building material or the design level of the predominant building class results in significant reductions in expected annual losses of physical damage, casualties and economic losses.

## 1 Introduction

Probabilistic seismic risk assessment methods provide a clear set of steps to estimate consequences of earthquakes such as physical damage to structures, economic losses and casualties. These estimates provide much needed quantitative information for decision making under uncertainty (Apostolakis, 2004; Ellingwood, 2005; Ellingwood and Kinali, 2009; Aven and Zio, 2011; Pasman and Reniers, 2014) to evaluate different mitigation strategies to reduce seismic risk (Liel, 2008; Cha and Ellingwood, 2013; Liel and Deierlein, 2013).

We develop a probabilistic seismic risk assessment model with an application to the coastal city of San Antonio, Chile. We implement a set of state-of-the-art hazard and exposure models, and we estimate risk curves for physical damage, economic losses and casualties. We generate a high-resolution model for the exposed building stock, considering the built area at a census block level and including different occupancy classes, namely residential, commerce, industry, etc. We use this model to analyze the implications of various counterfactuals involving changes in building materials and/or design levels of the



exposed building stock. We start the risk estimation process with the generation of a synthetic earthquake catalog, which
translates into loss distribution functions, from which usual risk metrics (e.g., expected annual loss, annual rate of exceedance, etc.) can be computed for different outcomes of interest to decision-makers, namely physical damage, economic losses and casualties. The method uses Monte Carlo simulation throughout the entire earthquake catalog, starting from an earthquake rupture scenario at a particular source defined by parameters such as the earthquake hypocenter and magnitude, and following it to its full realization, which include: i) rupture generation; ii) ground motion intensities; iii) damage to exposed buildings;
and iv) economic losses and casualties. Thus, the probabilistic approach implemented in this study, considers not only the analysis of one extreme seismic earthquake, but multiple possible scenarios through Monte Carlo simulations. The model's basic stages are explained in detail in the following section.

The proposed probabilistic risk tool is amenable to study and compare endless different counterfactual questions, e.g., How would the loss distribution change if the construction materials of the exposed buildings had been different?; or How an earlier
adoption of the seismic design codes would impact on the loss distribution?; or Would the number of injured people be smaller under a different population density?, etc. In this work, we use the model to quantitatively evaluate the impact of changes to particular building typologies, versus the current exposed building stock, from the perspective of different risk metrics, such that it can be used to better inform public policy decisions. For this purpose, several counterfactual analyses are performed, as to directly quantify the consequences for each countermeasure, using our proposed seismic probabilistic risk assessment
to ask: How would the loss, casualties, and physical damage distributions change, if the design code and/or materials of the exposed buildings had been different?.

A widely-used approach for seismic risk assessment is the Performance Based Earthquake Engineering (PBEE) methodology (Moehle and Deierlein, 2004), which follows a probabilistic approach and can be applied to evaluate seismic risk for a specific site and for a specific structure or building class. Meanwhile, to evaluate seismic risk for spatially distributed systems or
urban areas, the Monte Carlo Simulation approach (Musson, 2000) is the preferred method as it enables the straightforward incorporation of spatial correlations and uncertainty propagation (Crowley and Bommer, 2006; Jayaram and Baker, 2010; Poulos et al., 2017; Baker and Cornell, 2008), allowing to capture infrequent large loss scenarios, which have a significant impact on the tail of the risk distribution (Weatherill et al., 2015). Furthermore, when assessing risk in a specific building class, a single intensity measure can be used (Rodrigues et al., 2018; Goda et al., 2021). However, when different building
classes are considered, it is important to include the cross-correlation between the intensity measures at different sites (Loth and Baker, 2013). Moreover, spatial correlations not only between intensity measures, but also in damage between nearby structures should be included (Heresi and Miranda, 2022; Wang and Ellingwood, 2022), however, the lack of data regarding correlated damages makes it difficult to incorporate the latter. It is worth noting that a framework to extend PBEE to a regional scale was recently proposed, so it could be used in groups of structures spatially distributed (Heresi and Miranda, 2023).
One key aspect when assessing seismic risk is the exposure model, i.e., the physical and spatial characterization of elements, systems and people exposed to hazard. The detail of the available data often defines the resolution and methods by which exposure models are constructed. Usually, data at the country or regional scale is more accessible. Consequently, a top-down approach is employed to construct coarse resolution exposure models. This approach relies on aggregated national information,



such as Gross Domestic Product (GDP) or other economic indicators, which are then distributed onto a geographic grid using
spatial data as proxies (Silva et al., 2015; Villar-Vega et al., 2017). However, following a bottom-up approach, which consists on
creating a detailed exposure model, characterizing each building by its attributes and location, allows to create a representative
model of the area of study and a better appointment of it into a fragility model, enabling a better representation of their seismic
performance (Dabbeek et al., 2021). Another alternative is to create statistical exposure models based on a probabilistic nature
using the benefits from both approaches (Pittore and Wieland, 2013; Scheingraber and Käser, 2019). For the Chilean context,
Santa María et al. (2017) developed a residential national exposure model following the top-down approach based on the
population census and locally contrasted it with online surveys in three regional capital cities, while Gómez et al. (2022)
developed an exposure model following a statistical approach for the residential building stock of Valparaíso. Additionally,
Aguirre et al. (2018) followed a bottom-up approach to build a high resolution residential exposure model for Iquique.

Although significant research outcomes have been reached in the past decade regarding seismic risk assessment, including
assessments in Chile on a national scale (Marulanda et al., 2021; León et al., 2022; Marulanda et al., 2022), the novelty
of this contribution is mainly focused on three aspects: (i) the implementation of a probabilistic approach to risk analysis
at the urban scale based on Monte Carlo simulations and considering correlation parameters specifically developed for the
Chilean subduction zone; (ii) the risk quantification based on a detailed bottom-up exposure model at a high resolution level of
census block within a city, considering different building classes; (iii) the evaluation of different counterfactuals, changing the
predominant building classes of the exposed building stock to identify the best alternatives for mitigation plans including not
only physical damage, but also economic losses and casualties. It is important to remark that our focus is placed on people and
exposed buildings, considering not only the residential sector, but also other sectors such as industrial and commercial using
public available databases. In principle, the model is relatively easy to implement and replicate to any city or region for which
similar administrative data are available.

## 2   SEISMIC RISK ASSESSMENT METHODOLOGY

Risk consequences can be described by a loss exceedance curve, which shows the mean annual rate at which different levels
of losses or consequences are exceeded:

$$\lambda_C(c) = \nu \Pr(C > c),  \tag{1}$$

where $\Pr(C > c)$ is the probability of a consequence variable $C$ exceeds a threshold $c$, and $\nu$ is the mean annual rate of
significant earthquakes.

A useful metric to evaluate risk reduction measures is the expected annual loss ($E[C]$), which can be estimated as the area
under the loss exceedance curve (Baker et al., 2021), such that:

$$E[C] = \int_0^\infty \lambda_C(c)\,\mathrm{d}c  \tag{2}$$


Inspired in a Monte Carlo framework, we generate a synthetic earthquake catalog and Eq. (1) is solved simply by calculating the exceedance rates of a consequence variable from the drawing of numerous samples, with the following expression:

$$\lambda_C(c) \approx \frac{\nu}{N} \sum_{i=1}^{N} I(c_i > c), \tag{3}$$

where $N$ is the number of earthquake scenarios in the catalog, $c_i$ is the consequence value corresponding to scenario $i$, and $I(c_i > c)$ is the indicator function which equals 1 if $c_i > c$ and 0 otherwise. Similarly, Eq. (2) is computed as the mean of the loss multiplied by the mean annual rate of significant earthquakes, resulting in the following expression:

$$E[C] \approx \frac{\nu}{N} \sum_{i=1}^{N} c_i. \tag{4}$$

Figure 1 schematically shows the different stages involved to solve Eqs. (3) and (4). First we develop a high resolution exposure model, were we characterize building classes, people distribution and soil type for each census block in the city. Second, we develop a synthetic earthquake catalog containing $N$ seismic scenarios, where each one is defined by a magnitude and hypocenter location. Then, for each earthquake scenario in the catalog and for each block in the city, we estimate ground motion intensity measures (IM), seismic damage of every building class exposed, and economic losses and casualties produced by seismic damage. Finally, we integrate the results of all earthquake scenarios in the catalog to evaluate $\lambda_C$ and $E[C]$ from Eqs. (3) and (4), respectively.

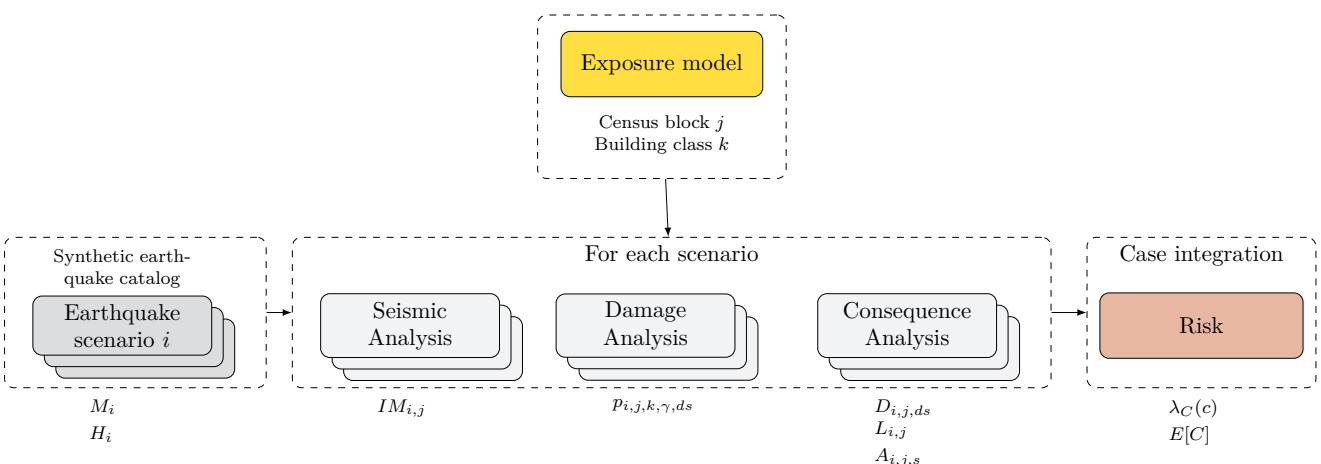

**Figure 1.** Stages in probabilistic seismic risk assessment methodology.

The following subsections provide necessary details of the methodology at each stage.



## 2.1 Exposure model

In the seismic risk assessment context, the exposure model refers to the compilation and characterization of all elements and systems (buildings, people, critical networks, etc.) existing in regions exposed to hazards, and therefore susceptible to damage or loss (Jaiswal et al., 2010). In order to develop a comprehensive and accurate risk assessment for the city of interest, we build the most detailed exposure model possible considering public databases, at a high resolution level of census block within the area of study. Thus, for each census block within the city, we consider detailed and geo-referenced information of: (i) exposed building stock; (ii) people distribution; and (iii) soil type.

The exposed building stock is characterized in terms of building classes, which define a group of structures that are assumed to posses a similar seismic response. We consider the building classes used by Hazus (FEMA, 2020) which are characterized by: (i) building typology defined by the construction material, structural system and building height; and (ii) seismic design level, which refers to the level of the design standards considered in the building design, and is defined primarily by the building's age of construction.

We characterize the exposed building stock based on the information from a property cadaster provided by the Chilean Internal Revenue Service (Servicio de Impuestos Internos, SII). This database (hereinafter called SII database) contains detailed information of each construction unit within a real state property, including address, year of construction, occupancy class, construction material, construction quality, and built area, among others. We use this information to assign a building typology and seismic design level to each construction unit, which in turn is geo-referenced and assigned to a census block.

To assign a building typology to each construction unit, we first consider the material and occupancy class from the SII database, as summarized in Table 1 for the most common cases. For example, any construction unit made with masonry or wood is assigned the RM1 or W1 building type, respectively, regardless of its occupancy class. For concrete or steel units, the building type is assigned based on the occupancy class since, for example, residential structures in Chile are typically built using shear walls or braced frames, respectively. Then, we estimate the building height by analyzing the apartment numbers listed in the address of each construction unit within the database. We classify each building type according to its number of stories as low-rise (L) for 1 to 3 stories, mid-rise (M) for 4 to 7 stories, or high-rise (H) for 8 or more stories.

**Table 1.** Building typology assignment

| Material | Occupancy class | Building Typology | Description |
|----------|-----------------|-------------------|-------------|
| Masonry | All occupancies | RM1 | Reinforced masonry bearing walls |
| Wood | All occupancies | W1 | Wood, light frame |
| Concrete | Residential, hotel or education | C2 | Concrete shear walls |
| Concrete | Remaining occupancies | C1 | Concrete moment frame |
| Steel | Residential, hotel, education or warehouse | S2 | Steel braced frame |
| Steel | Remaining occupancies | S1 | Steel moment frame |



Finally, we assign the seismic design level defined by Hazus (FEMA, 2020): pre-code (PC), low-code (LC), moderate-code (MC), or high-code (HC), to each construction unit based on its quality and year of construction, both available directly from the SII database. According to the Chilean building standard and practice, the first construction regulation was published in 1935 under the General Construction and Urbanization Law (Minvu, 1935). Later, the first seismic standard was introduced in 1972 (INN, 1972) and then updated in 1996 (INN, 1996). Thus, the seismic design level was assigned based on the year of construction according to this age distribution, as shown in Table 2. Moreover, the quality of construction is also considered, which includes additional attributes such as the quality of structural elements and building finishes, as well as identifying defective components or design (SII, 2018). Thus, independently of the year of construction, buildings with a quality of construction of 4 or 5 are assigned to design level LC and PC, respectively.

**Table 2.** Design level description.

| Year of construction | Design level |
|---|---|
| < 1935 | PC |
| [1935 - 1972) | LC |
| [1972 - 1996) | MC |
| ≥1996 | HC |

Thus, the building class $k$ is defined by the building typology (Table 1), the building height (L, M or H), and the design level (Table 2). As a result of this process, we obtain the built area $a_{k,j}$ for each building class $k$ within each census block $j$.

We estimate the economic value as the monetary value of each construction unit. Following the methodology proposed by SII (2018), it corresponds to the building construction cost as estimated by its size in squared meters, material and construction quality, then depreciated according to its age at a constant yearly depreciation rate whose value depends on the construction material. The advantage of using this methodology over, say, market prices, is that it excludes the land value, keeping only the building's economic value. Thus, the economic value $v_{k,j}$ of a construction unit of building class $k$ in census block $j$ is calculated as:

$$v_{k,j} = a_{k,j} \cdot cc_k \cdot d_k, \tag{5}$$

where $a_{k,j}$ is the built area in square meters; $cc_k$ is the unit construction cost per square meter for building class $k$, depending on the building type, material and quality of construction; and $d_k$ is a depreciation factor that depends on the year of construction and the material of the building class $k$.

In addition, we use data from the Chilean census provided by the National Statistics Institute (INE) to determine the population distribution by census block. To estimate the number of people associated with each construction unit within a census block, we allocate the population proportionally based on the built area of each unit with residential occupancy. This population distribution is later used to estimate the potential number of casualties resulting from building physical damage. As such, we define $n_{k,j}$ as the number of inhabitants assigned to a construction unit of building class $k$ within census block $j$.





155     Furthermore, to estimate the seismic intensities at each census block, it is necessary to characterize the local site conditions. In this study, we use the mean shear wave velocity in the upper 30 meters ($V_{s30}$), which is a parameter adopted by the latest ground motion models (GMMs) that are discussed in Section 2.3. To determine the $V_{s30}$ value for each census block, we use publicly available information from the SIGAS microzonation project (Sernageomin, 2020), which integrates various data sources and assigns a seismic soil type to different areas along the Chilean coast. Based on this classification, we assign a

160  soil type to each census block according to its location. Finally, we assign a $V_{s30}$ value to each seismic soil type based on the average value of the interval limits defined by the Chilean seismic standard DS61 (Minvu, 2011), except for soil type A, which has a fixed value of 1000 m/s. The $V_{s30}$ values considered for each seismic soil type are summarized in Table 3.

**Table 3.** Seismic soil types.

| Soil type | Description | $V_{s30}$ (m/s) |
|---|---|---|
| A | Rock, cemented soil | 1000 |
| B | Soft rock or cracked, very dense or firm soil | 700 |
| C | Dense or firm soil | 425 |
| D | Moderately dense or firm soil | 265 |
| E | Medium compact soil | 90 |

## 2.2   Synthetic earthquake catalog

We employ Monte Carlo simulations to generate a synthetic earthquake catalog by sampling earthquake hypocenter locations

165  $H_i$ and moment magnitudes $M_i$, in line with the Gutenberg-Richter relation (Gutenberg and Richter, 1944). We use the recurrence model developed by Poulos et al. (2019), the latest model to simulate interface earthquakes-rate production along the Chilean central subduction zone. The mean annual rate of significant earthquakes is calculated as $\nu = 10^{a-b \cdot M_{\min}}$, using earthquake recurrence parameters $a$ and $b$ corresponding to subduction zone 2 in the zonation model (Poulos et al., 2019). To determine the location of each earthquake scenario, we sample earthquake hypocenters from a spatial uniform distribution

170  within the subduction interface region, as defined by the Slab2 geometry model (Hayes et al., 2018). This approach enables us to generate a comprehensive catalog of synthetic earthquakes for our seismic risk assessment method.

    We use the earthquake source-scaling law proposed by Goda et al. (2016) to simulate the spatial rupture dimensions across the fault for each synthetic earthquake, based on its magnitude. This model was selected because it is constrained with a much larger magnitude range, including magnitudes smaller than $M_w 7.1$ (Allen and Hayes, 2017). Using this scaling relationship, we

175  determine the fault length and width. Once the rupture area is defined, we use the Slab2 model (Hayes et al., 2018) to constrain the rupture surface based on the subduction fault interface geometry. We define this rupture surface in order to identify the closest distance from the site of interest to the rupture, which is used later to estimate IMs (Der Kiureghian and Ang, 1977).



## 2.3 Seismic analysis

The seismic energy released during an earthquake propagates from the ruptured fault to each site of interest, resulting in
a diverse range of intensity measure values ($IM_{i,j}$) across census blocks. These values depend on several factors such as
earthquake magnitude, the location of the ruptured fault with respect to the sites of interest, local site conditions, and other
sources of uncertainty, which can be modeled as random variables to account for both stochastic and epistemic uncertainties.

The intensity measures can be obtained from simulations of seismic wave propagation (Zhang et al., 2023). However, com-
putational costs limit the number of scenarios that can be simulated. When thousands of simulations are required, ground
motion models (GMMs) —also known as ground motion prediction equations— offer a highly efficient solution to estimate
ground motion intensities at a particular site due to an earthquake of a specific distance and magnitude. The IMs predicted
are typically peak ground acceleration (PGA), velocity, or displacement, and spectral acceleration at different periods. In this
study, we consider three IMs which correspond to spectral acceleration at different periods: PGA, $S_a(0.3\,\text{s})$, and $S_a(1.0\,\text{s})$.
They are used to define the demand spectrum in each earthquake scenario, as discussed later in Section 2.4.

GMMs are typically based on statistical analyses of observed ground motion records from past earthquakes, along with other
relevant geological and seismological factors (Douglas, 2003). They typically provide an estimate of the median level of IM
and its uncertainty at a given site. To improve the accuracy of the estimates, they can also take into account spectral and spatial
correlations in their residual terms, resulting in a more constrained and realistic characterization of the variability of IMs at a
specific location. By incorporating these correlations, the GMMs can better account for the complex interactions between the
earthquake source, the propagation path, and the local site effects that influence the ground motion at the site of interest.

We implement Candia et al. (2020)'s GMM, which is based on the same functional form than BC Hydro's model (Abra-
hamson et al., 2016), but with recalibrated parameters and spectral correlation coefficients that were constrained with observed
ground motions recorded on the Chilean subduction zone. Furthermore, we include Heresi and Miranda (2019) spatial correla-
tions to address the spatial variability of IMs. This was preferred as it demonstrated stronger correlations at short distances, as
compared to alternative models (Jayaram and Baker, 2009; Goda and Atkinson, 2010), which is more adequate to model IMs
from subduction earthquakes. Additionally, local site conditions are considered in the GMM by incorporating the influence of
soil conditions at the sites of interest trough the $V_{s30}$.

The result of this stage corresponds to spatially correlated IM maps, consistent with local $V_{s30}$ conditions, for each earth-
quake scenario in the catalog ($IM_{i,j}$).

## 2.4 Damage analysis

Damage analysis involves estimating the probability of a specific building class $k$ being in a particular damage state based on
a given IM at a specific site $j$ and for a given scenario $i$, $IM_{i,j}$. In this study, we implement the Capacity Spectrum Method as
described by Porter (2009). This method determines the performance point, i.e., the intersection between the capacity spectrum
of building class $k$ and the demand spectrum of the scenario $i$ and site $j$. The performance point is then associated with a specific
damage state using a fragility function.


On the one hand, the capacity spectrum is defined by representing the building class as a single-degree-of-freedom nonlinear damped oscillator (Porter, 2009). The capacity spectrum corresponds to the pushover curve in terms of the spectral acceleration capacity versus spectral displacement of a structure, and it is divided into three parts: linear, up to the yield point $(D_{\mathrm{y}}, A_{\mathrm{y}})$, perfectly plastic beyond the ultimate point $(D_{\mathrm{u}}, A_{\mathrm{u}})$, and an elliptical spline in between, as shown in Figure 2(a). For each

building class included in the exposure model, we use the capacity spectrum provided by Hazus (FEMA, 2020).

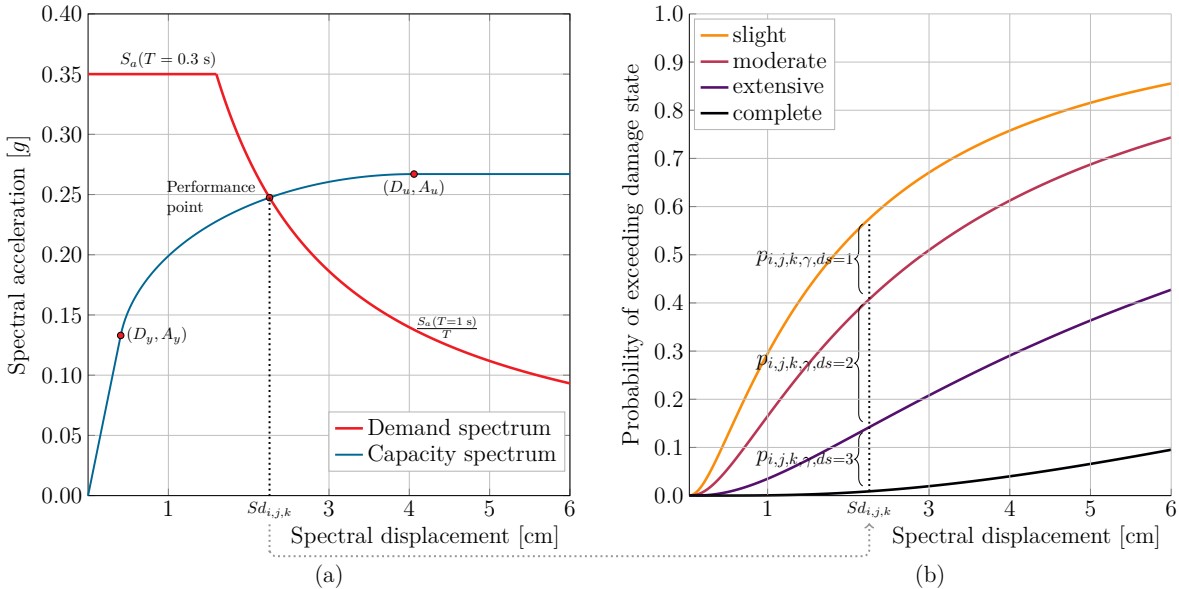

**Figure 2.** (a) Demand spectrum for seismic scenario $i$ census block $j$, and capacity spectrum for building class $k$ ; and (b) fragility curves for a specific building class, physical component and damage state.

On the other hand, the seismic demand spectrum is based on the seismic intensity measures $S_a(0.3\ \mathrm{s})$ and $S_a(1\ \mathrm{s})$, and corresponds to an idealized elastic response spectrum for 5% damping (Porter, 2009). The input spectrum is defined mainly in two spectral zones: constant acceleration and constant velocity, and is modified by reduction factors to account for damping ratios other than 5% to obtain the demand spectrum schematically shown in Figure 2(a).

The point at which the capacity spectrum intersects with the demand spectrum defines the performance point $Sd_{i,j,k}$ for each seismic scenario $i$, census block $j$ and building class $k$, as shown in Figure 2(a). Then the fragility function is used to estimate the probability that each of the building components is in each of several damage states given a certain performance point $Sd_{i,j,k}$. Four damage states $(ds)$ are considered: slight, moderate, extensive and complete, numbered from 1 to 4, respectively. Additionally, three building components $(\gamma)$ are considered: structural components, nonstructural components sensitive to

acceleration, and nonstructural components sensitive to drift, numbered from 1 to 3, respectively. We use the fragility functions according to the definitions provided by Hazus (FEMA, 2020), which are defined for each building class, damage state and building component as log-normal functions. In Figure 2(b), fragility curves for a particular building component and building




class are shown schematically. As a result of this stage, the damage state probabilities $p_{i,j,k,\gamma,ds}$ are obtained, which correspond to the probability of being at a damage state $ds$ given a building component $\gamma$, building class $k$, census block $j$, and seismic

scenario $i$.

## 2.5 Consequences analysis

In consequences analysis, the damage state probabilities calculated earlier are utilized to evaluate different variables of interest for stakeholders. In this study, we focus on three consequence metrics: physical damage to the building's structural components, direct economic losses, and casualties. These metrics are determined for each seismic scenario $i$ and census block $j$.

We calculate the physical damage $D_{i,j,ds}$ on the structural components ($\gamma = 1$) for each damage state $ds$ as:

$$D_{i,j,ds} = \frac{\sum_{k=1}^{K} a_{k,j} \cdot p_{i,j,k,\gamma=1,ds}}{\sum_{k=1}^{K} a_{k,j}}, \tag{6}$$

where $p_{i,j,k,\gamma=1,ds}$ is the probability of being on each damage state $ds$, and $a_{k,j}$ is the built area of each building class $k$.

Additionally, we calculate the direct economic losses $L_{i,j}$ of a census block $j$ for the seismic scenario $i$ as:

$$L_{i,j} = \sum_{k=1}^{K} v_{k,j} \sum_{\gamma=1}^{3} \sum_{ds=1}^{4} p_{i,j,k,\gamma,ds} \cdot r_{k,\gamma,ds}, \tag{7}$$

where $v_{k,j}$ is the economic value of a construction unit of building class $k$ at census block $j$, $p_{i,j,k,\gamma,ds}$ is the probability that each building component $\gamma$ is in a damage state $ds$, and $r_{k,\gamma,ds}$ is the building repair and replacement ratio, which represents the ratio of the total cost of repairing a building to the total cost of replacing it with a new one and is obtained from Hazus (FEMA, 2020) for each building class, building component and damage level.

Finally, we calculate the casualties $A_{i,j,s}$ with severity level $s$ as the count of individuals who are injured as a result of
an earthquake. We consider four severity levels: light injuries ($s = 1$), hospitalized injuries ($s = 2$), life threatening injuries ($s = 3$), and deaths ($s = 4$), as defined by Hazus (FEMA, 2020). For each severity level $s$, we assess the estimated number of casualties as:

$$A_{i,j,s} = \sum_{k=1}^{K} \sum_{ds=1}^{4} \frac{a_{k,j}^{r} \cdot n_{k,j}}{\sum_{k=1}^{K} a_{k,j}^{r}} \cdot p_{i,j,k,\gamma=1,ds} \cdot f_{k,ds,s}, \tag{8}$$

where $a_{k,j}^{r}$ is the built area of building class $k$ in census block $j$ with residential occupancy, $n_{k,j}$ is the number of inhabitants
located in a construction unit of building class $k$ at census block $j$, $p_{i,j,k,\gamma=1,ds}$ is the probability of being in a damage state $ds$ for construction unit $k$ and building component $\gamma = 1$, and $f_{k,ds,s}$ is the casualty rate for a severity level $s$, provided by Hazus (FEMA, 2020) for each building class $k$ and damage level $ds$.



# 3  CASE STUDY SAN ANTONIO CITY

In this section we provide an overview of the baseline, followed by a description of the counterfactuals, which will be evaluated
and compared later in Section 4.

## 3.1  Baseline

Based on the current available data, San Antonio has a total built area of $2\,591\,156$ m$^2$ and a total population of $80\,322$
inhabitants. Figure 3(a) shows that residential is the predominant occupancy class, representing 65% of the total built area,
followed by warehouse (11%), commerce (9%), education (5%), office (3%), and industry (2%). The most commonly used
construction materials are masonry, wood, steel, and concrete, which account for 41%, 31%, 14%, and 10% of the built area,
respectively, as depicted in Figure 3(b). Additionally, Figure 3(c) illustrates the distribution of buildings in terms of their year
of construction. It shows that 45% of the total built area consists of buildings constructed after 1996, followed by 28% built
between 1972 and 1996, 23% between 1935 and 1972, and only 3% before 1935. Moreover, nearly 90% of the buildings in the
study area correspond to low-rise structures.


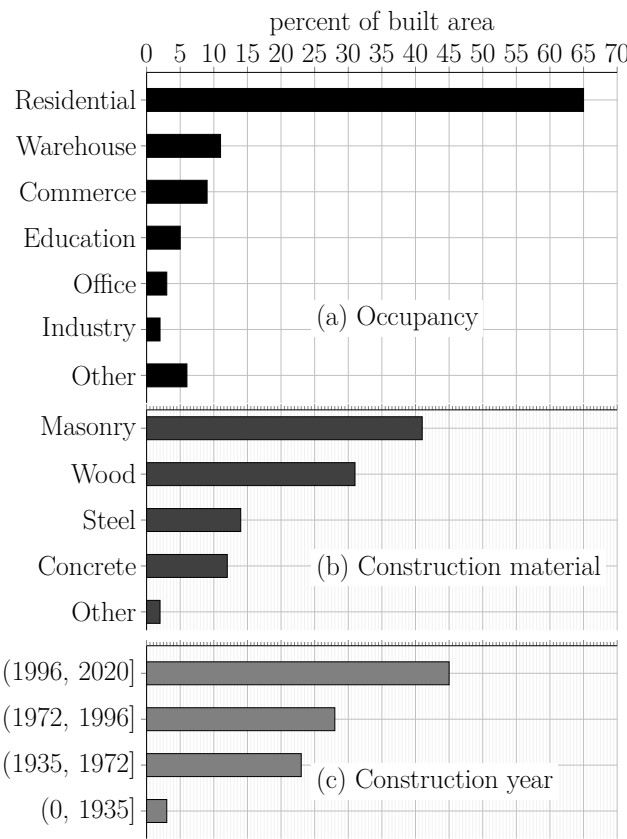

**Figure 3.** San Antonio built environment.

The distribution of building typology and seismic design level are shown in Figure 4 in terms of built area, economic value, and population. It is evident from the figure that the two most common building classes in San Antonio are RM1L-LC and W1-LC, with 27% and 21% of the total built area, respectively. These building classes also contribute to 38% of the total economic value, and 72% of the total population resides in buildings of these classes.


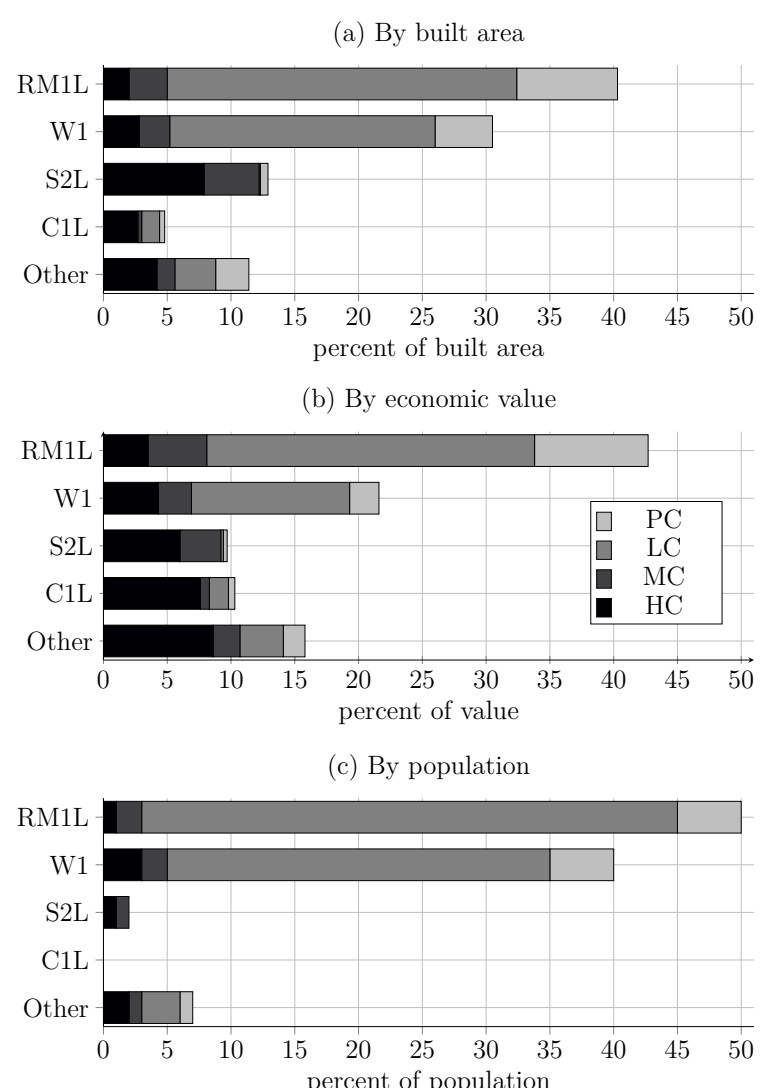

**Figure 4.** Building typology and design level.

The spatial distribution of the exposed building stock can be appreciated in Figure 5, which shows the predominant occu-
pancy classes, building typologies, and percentage of economic value per census block, respectively. The figure reveals that
the most common building typologies, RM1L and W1, are primarily located in residential areas with low economic value.
Additionally, the areas with highest economic value correspond to commercial and warehouse areas, which correspond to the
port zone.


Furthermore, Figure 6 illustrates the spatial distribution of the normalized population density (a) and soil type (b), indicating
that most of the residential areas are situated in soil types B and C, while the commercial and port areas are located in lower
quality soils.

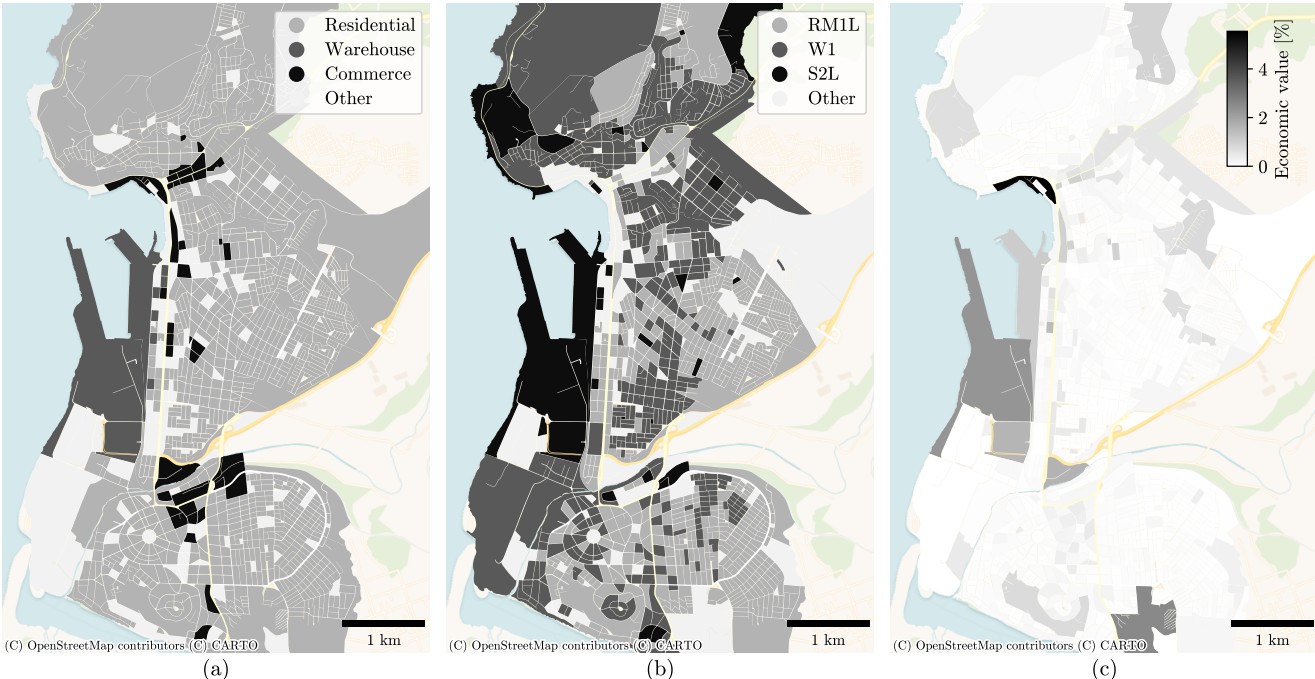

**Figure 5.** (a) Predominant occupancy classes, (b) predominant building classes, and (c) economic value per census block.





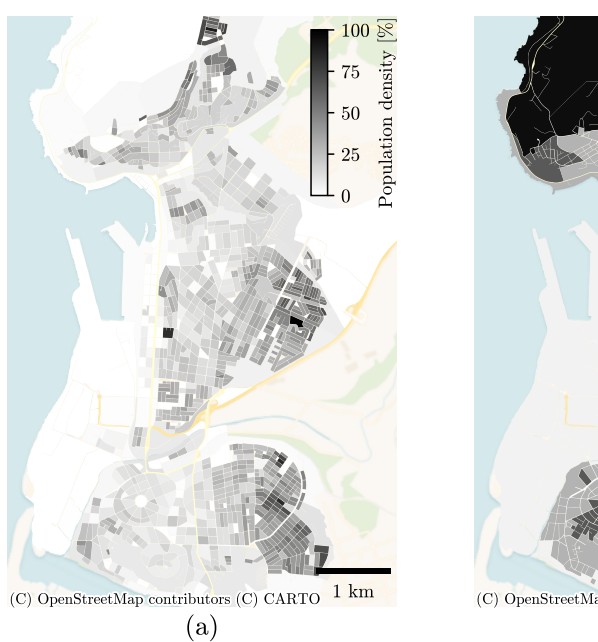

**Figure 6.** (a) Normalized population density distribution and (b) soil type distribution.

## 3.2 Counterfactuals

We analyze four counterfactuals, each modifying the exposed building stock in a particular way, so as to assess its impact on the main risk metrics. As previously shown in Figure 4, the building classes RM1L-LC and W1-LC are the most prevalent in San Antonio. Therefore, the counterfactuals focus on changing these building classes in terms of building material and/or seismic design level. CF1 involves modifying the seismic design level of the predominant building type RM1L from LC to HC, resulting in a significant enhancement of its structural performance. In CF2, the predominant building class RM1L-LC is replaced with C2L-HC, not only changing the design level but also the construction material from masonry to reinforced concrete. Similarly, CF3 focuses on improving the design level of the second most prevalent building type W1 from LC to HC. Finally, CF4 replaces the wooden buildings with typology W1-LC with reinforced concrete buildings with typology C2L-HC. The counterfactual definitions are summarized in Table 4.

Any of these changes would have meant a greater economic investment at the time of construction, resulting in higher valuations today. The difference in total economic value with respect to the baseline is shown in Table 4. The table shows that modifying only the design level (cases CF1 and CF3) results in increases in economic value of 17% and 12% for masonry and wood, respectively, while changing both material and design level to reinforced concrete C2L-HC, leads to a significantly higher value change of 29% in both cases (CF2 and CF4). Figure 7 shows the percentage of built area of each building typology and design level modified in each counterfactual, as well as the percentage of built area in the baseline.





**Table 4.** Counterfactual definitions.

| Counterfactual | Definition | Value change |
|---|---|---|
| Baseline | Current exposure | Current value |
| CF1 | RM1L-LC to RM1L-HC | 17% |
| CF2 | RM1L-LC to C2L-HC | 29% |
| CF3 | W1-LC to W1-HC | 12% |
| CF4 | W1-LC to C2L-HC | 29% |

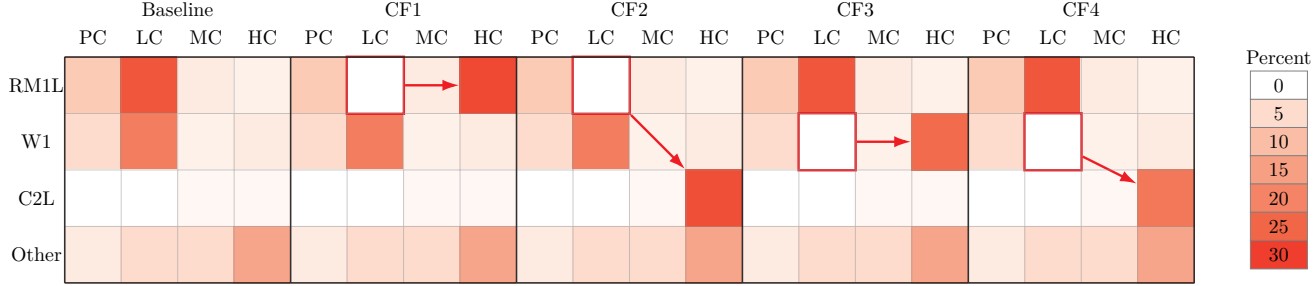

**Figure 7.** Counterfactual definitions: percent of built area in each typology and design level.

Figure 8 shows fragility functions for the predominant building classes in the base scenario RM1L-LC and W1-LC, together with the building classes considered in the counterfactuals RM1L-HC, W1-HC and C2L-HC. Fragility functions are obtained
from Hazus database (FEMA, 2020), and show the probability that structural building components will exceed each of the damage states given a specific spectral displacement. It can be observed that for a given spectral displacement level, the probability of experiencing slight damage is similar for all building classes. However, as the damage state increases, the fragility functions begin to diverge, and the probability of incurring into extensive or complete damage is substantially lower for C2L-HC than it is for RM1L-LC and W1-LC at the same displacement. However, it is important to notice that the spectral
displacement corresponds to the performance point $Sd_{i,j,k}$ for each seismic scenario $i$, census block $j$ and building class $k$, thus the input for the fragility function may not be the same displacement for different building classes and census blocks, even for the same seismic scenario $i$.

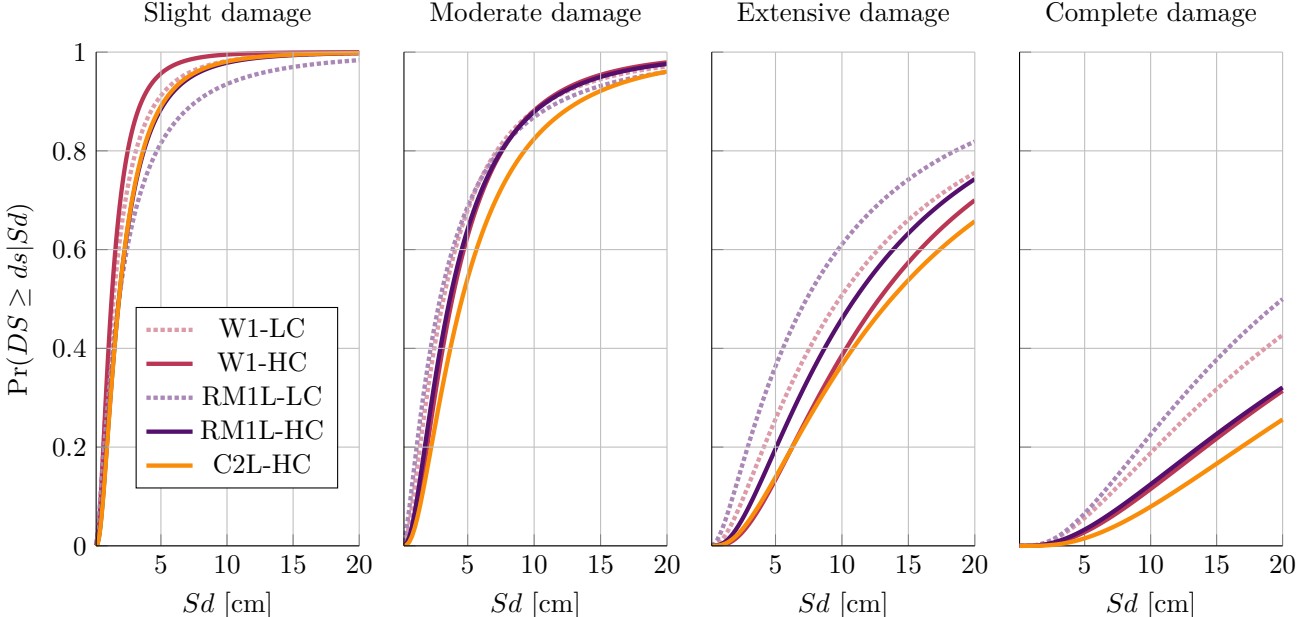

**Figure 8.** Fragility curves.

## 4 COUNTERFACTUAL RISK ANALYSIS

To calculate the risk metrics using the methodology proposed in Section 2, we generate a synthetic seismic catalog consisting
of 60 000 earthquake scenarios with magnitudes ranging from $M_w$ 6 to 9 and hypocenters spatially distributed uniformly, as
shown in Figures 9(a) and (b), respectively. It is evident from Figure 9(a) that there are many scenarios with low magnitudes, but
as the magnitude increases, the frequency decreases. Additionally, each point in Figure 9(b), represents hypocenter locations
$H_i$ and moment magnitudes $M_i$ of each earthquake scenario $i$ in the catalog, located within subduction Zone 2 as defined by
Poulos et al. (2019). The location of San Antonio City is also indicated in the Figure with a white square. For each scenario
in the catalog, we sequentially conduct seismic, damage and consequence analyses (Figure 1). Finally, we compute the mean
annual rate of exceedance $\lambda_C$ and the expected annual loss $E[C]$ for the main consequence variables namely physical damage,
direct economic losses, and casualties, using Eqs. (3) and (4), respectively.

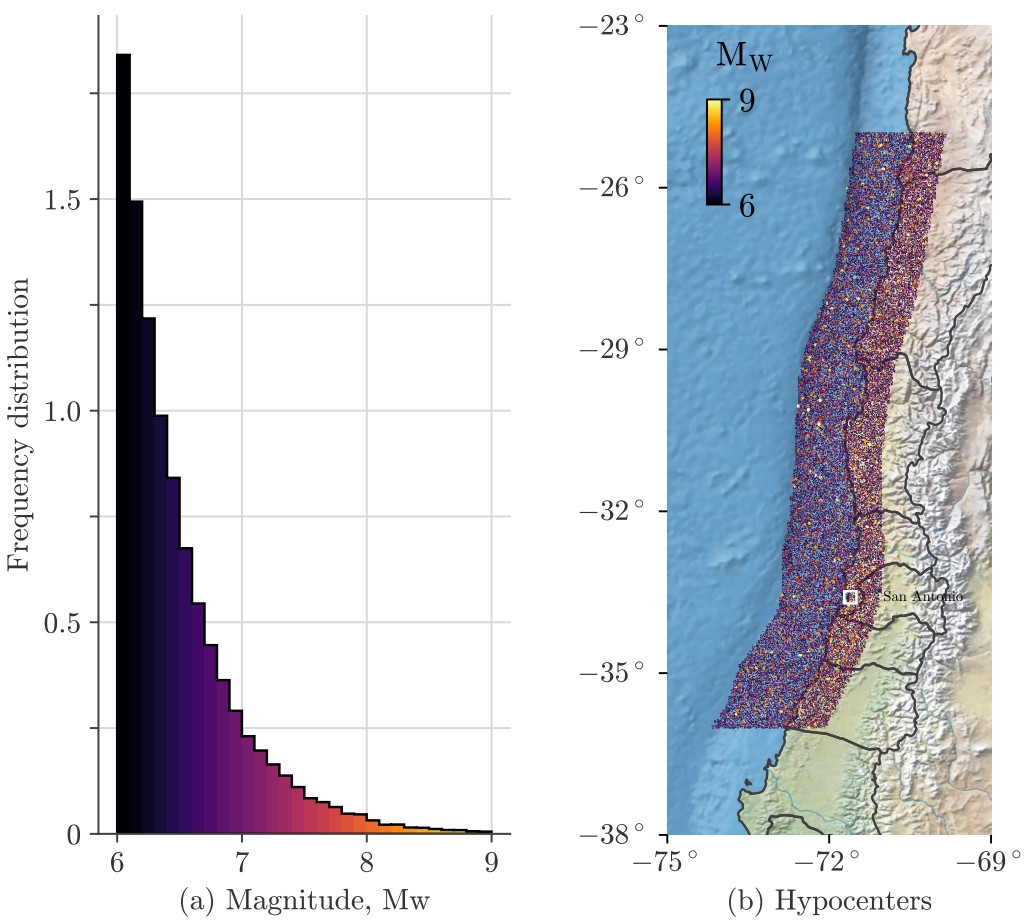

**Figure 9.** Earthquake scenarios. (a) Magnitudes distribution and (b) Hypocenters location
The white square marks San Antonio.

Risk curves for the main consequence variables are presented in Figure 10 for the baseline, alongside those of the four counterfactuals. Figure 10a shows the mean annual rate of exceedance of the percentage of total built area that is affected by physical damage to structural components with complete damage ($ds4$). It is evident that counterfactuals CF1 and CF2 have the most significant impact on this consequence variable, leading to a substantial reduction in the percentage of built area with complete damage. Negligible difference can be inferred from this result between cases CF1 and CF2. On the other hand, cases CF3 and CF4 have a lesser impact on the risk curve. These results imply that modifying the most predominant building class RM1L-LC, either in the design level (CF1) or both in the material type and design level (CF2), would be desirable to reduce physical damage in the city. This occurs because of the large built area of this typology in San Antonio (Figure 7), and the significant improvement in the fragility that is achieved with the proposed changes, as shown previously in Figure 8. Additionally, changes in the second predominant building class W1-LC, either in the design level (CF3) or both in the material





type and design level (CF4), would also reduce the seismic risk of the city in terms of built area with complete damage, although to a lesser extent.

These results mean that, for the case study city, despite the original exposed built area of the two predominant building typologies RM1L-LC and W1-LC being similar (27% and 21%, respectively), changing masonry in either design level or construction material has a larger impact than changes on wood, mainly due to the lower seismic fragility of wood compared to masonry (Figure 8). However, these results could be different if other hazards, such as tsunami or fire, were considered (Suppasri et al., 2013). The advantage of the methodology proposed in this study is that it allows for easy incorporation

of other hazards. For example, tsunami analysis can be implemented, even considering earthquake and tsunami sequential analysis consistent with the same source (Cortez et al., 2022). Additionally, climatic hazards such as floods from extreme precipitations (Bowers et al., 2022), can also be included within the same framework.

On the other hand, Figure 10b illustrates the mean annual rate of exceedance of direct economic losses. The percentage of loss is calculated as the total direct economic loss of each case relative to the total economic value of the baseline, resulting in

normalization to the same value for all cases. Once again, CF1 and CF2 demonstrate significant reduction in economic losses compared to the baseline, with CF1 exhibiting a higher reduction than CF2. This can be attributed to the fact that enhancing the design level (CF1) raises the total exposed value but to a lesser extent than in the CF2 case, as shown in Table 4, which translates into larger losses in CF2 for the same extent of physical damage than CF1. Additionally, case CF3 shows almost negligible reduction in economic losses, while CF4 produces higher economic losses than the baseline. This occurs because

the reduction of physical damage does not compensate the increase of the exposed economic value in this case.

Finally, Figure 10c, shows the mean annual rate of exceedance of the casualties with severity level 4. Once again, cases CF1 and CF2 exhibit a substantial impact on the risk curve, as they involve modifications in RM1L-LC, either in design level (CF1) or both material and design level (CF2). The reduction in casualties is primarily due to the decrease in the percentage of built area with complete damage shown in Figure 10a. In the case of CF3, the improvement in the design level of W1 has

an almost insignificant impact on casualties. However, in the case of CF4, the change from W1-LC to C2L-HC increases the risk of casualties with severity level 4 for low annual rates of exceedance. This is because, despite the decrease in the risk of complete damage (Figure 10a), the probability of casualties with severity level 4 given complete damage (factor $f_{k,s,ds}$ in Eq. (8)) is much higher for C2L than for W1, leading to a greater risk of casualties.



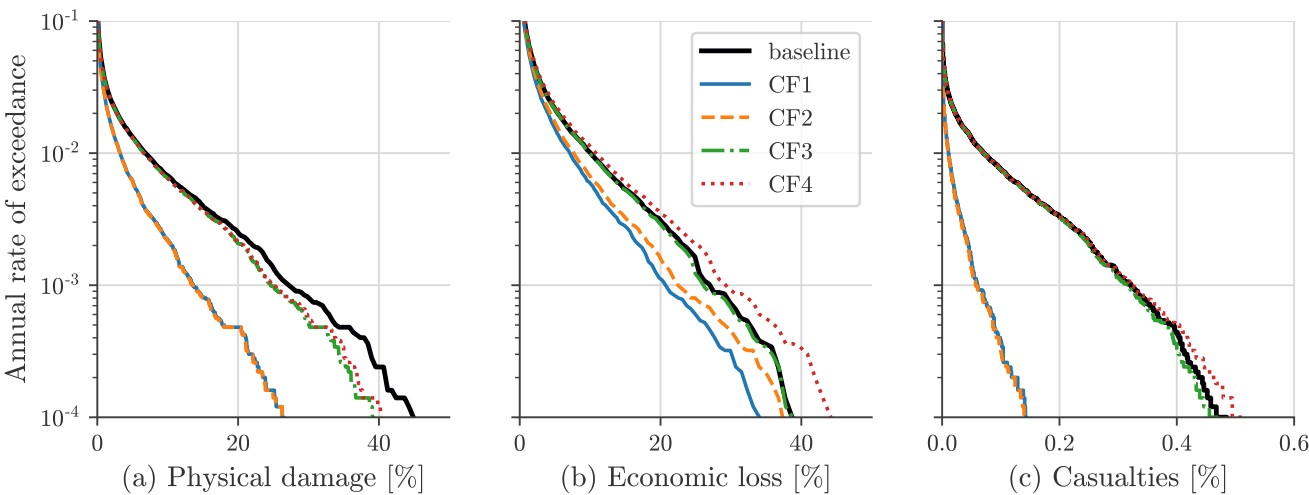

**Figure 10.** Risk curves comparison: (a) Built area with complete physical damage ($ds4$); (b) economic losses; and (c) casualties (severity 4).

Figure 11 shows the difference between the expected annual losses calculated using Eq. (4), for the same consequence
variables presented in Figure 10 between the baseline and counterfactuals. Once again, results reveal that cases CF1 and
CF2, i.e., changing RM1L-LC to either RM1L-HC or C2L-HC, has the most significant impact on consequence variables.
Furthermore, results demonstrate that modifying the design level (CF1) or building typology and design level (CF2) has an
almost identical effect in terms of reducing physical damage (complete), with reductions of more than 45% in both cases.
Additionally, this reduction in physical damage translates to almost identical reductions in casualties (severity level 4) of more
than 80% relative to the base scenario in both cases. In terms of economic losses, case CF1 results in a more significant
difference compared to the base scenario (24%) than CF2 case (17%). This is because modifying the material to reinforced
concrete (CF2) increases the building's economic value more significantly than modifying only the design level of masonry
(CF1), as can be inferred from Table 4. The figure also shows that improving the design level of W1 from LC to HC (CF3) has
minimal impact on reducing physical damage, and almost negligible effect on casualties and economic losses, as compared
to counterfactuals CF1 and CF2. The case CF4 produces the minimum reduction in physical damage (5%), and the expected
annual economic losses are larger than in the base scenario, indicating that this counterfactual worsens the conditions and
increases the risk of the city. The same is observed for casualties, although to a lesser extent.

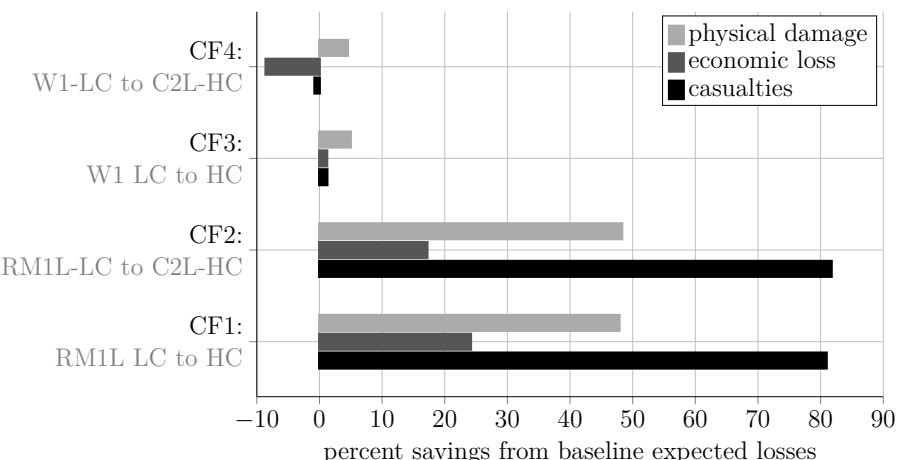

**Figure 11.** Counterfactual analysis.

## 5 Conclusions

A probabilistic seismic risk assessment model was proposed and applied to the city of San Antonio in Chile. Risk curves for
physical damage, casualties, and economic losses were obtained for the current exposed building stock and for four counter-
factuals. The exposure model was developed at a high resolution level of census block, allowing for buildings exposed to be
easily modified for evaluating different counterfactuals involving changes in building materials and/or design levels. Using
Monte Carlo simulations, seismic risk assessment involved generating a synthetic earthquake catalog and conducting seismic,
damage, and consequence analyses for each scenario in the catalog.

The predominant building class observed in the baseline was RM1L-LC, i.e, low rise masonry buildings with low design
level (27% of the total built area), followed by W1-LC i.e, low rise wood buildings also with low design level (21% of the
total built area). Results demonstrate that changing either the design level or the building material of the predominant building
class would significantly reduce the city's risk. Specifically, in terms of expected annual losses of the built area with complete
damage, the reduction obtained is roughly 48% compared to the baseline, leading to a substantial decrease in casualties of over
80%. However, when considering direct economic losses, the reduction achieved by changing the design level (24%) is greater
than the reduction from changing both construction material and design level (17%) due to the increased economic value of the
exposed building stock in the latter case. In contrast, changing either the design level or the construction material of the second
predominant building class W1-LC has a negligible impact on risk metrics and in some cases increases the city's risk.

Overall, our results emphasize the significant impact of material and design levels of the exposed building stock on the
consequences of seismic events. Although the outcomes are influenced by the capacity and fragility curves used in this study,
the methodology allows to quantitatively evaluate different risk metrics in various counterfactuals to make better informed
decisions. For example, the temporal population distribution and/or the urban expansion or densification processes, which are
key questions for urban planning.



Finnaly, by using Monte Carlo simulations, the proposed methodology allows consideration of spatial and spectral corre-

lations of seismic intensities and is easily extendable to other locations and even to other hazards, such as tsunami or floods. This flexibility enhances its utility in providing quantitative metrics for decision-makers to make informed decisions to reduce overall risk, making it a useful tool for public policy planning.

*Author contributions.* R. Jünemann and A. Urrutia conceived, conceptualized, supervised the research work and organized and edited the manuscript. A. Urrutia prepared the required data, implemented the risk model, run the simulations, and prepared the outputs and figures. M.

Damian compiled the required data, and collaborated in the implementation of seismic and damage analysis modules. O. Ortiz developed the exposure model and economic valuation. F. Zurita contributed to the economic valuation, consequence analysis and counterfactual definition, and also prepared figures. J. Crempien contributed in the development of the earthquake catalog and seismic analysis module. All co-authors participated in the analysis of the results and discussions, and contributed to the edition of the final manuscript.

*Competing interests.* The authors have no competing interests to declare.

*Acknowledgements.* The work included in this article was supported by the National Agency of Research and Development (ANID) under the grant FONDEF IT20I0054; FONDEF IT21I049 and Research Center for Integrated Disaster Risk Management (CIGIDEN), ANID/-FONDAP/1522A0005. We are also grateful to Omar Bello, who collaborated in the early stages of the project.



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
