# Peer review of "Probabilistic seismic risk assessment for cities: Counterfactual analysis in a Chilean case study"

_Natural Hazards and Earth System Sciences, 2023_

## Referee Comment (RC2)

Dear Editors NHESS,

Dear Rosita, Alejandro, Monserrat, Oscar, Felipe and Jorge,

I believe that the content of this paper is pertinent for publication in NHESS. Some of the results may have implications for disaster risk management. However, due to the disorganized structure of the paper and imprecisions, significant revisions are necessary to proceed with the next steps in publishing this work.

In terms of scientific merit, the article does not introduce new methods for seismic risk assessment. Novelty is a primary criterion for a manuscript to be published in a scientific journal. Although the article lacks novelty, it does present unique results on the seismic risk faced by the San Antonio study area in Chile, an important economic center in the country. However, these results need to be presented in a different manner to convincingly convey their relevance to readers.

I have some comments that may help enhance the quality of the article. If the authors accept my suggestions, profound modifications in the paper's structure will be necessary. I hope that responses to most of my comments can be incorporated into a new version of this paper. If they are not, in my opinion, the study should not be considered for publication in NHESS.

1.      MAJOR STRUCTURAL COMMENTS.

1.1.    Software and data availability

1.1.1.    Naming the software, referencing it. As a reviewer, I am naturally interested in examining the proposed "tool" mentioned by the authors in the paper, along with its advantages and limitations in comparison to other existing tools that may offer similar capabilities. Surprisingly, the name of this tool is not provided in the submitted manuscript, and I am curious about the reason behind this omission.

In my investigation, I visited the author's affiliation websites and discovered that the authors are currently involved in the Disaster Research Workflow (DRW) project. The project is described as "a computational platform focused on the simulation of natural hazards, such as earthquakes and tsunamis, and their possible consequences." I suspect that DRW is the tool referred to by the authors. Therefore, I am inquiring whether there are any conflicts preventing its mention and, more importantly, referencing in the text. I kindly request the authors to provide an explanation for this significant omission.

Regardless of whether DRW is the actual name of the tool or not, the specific name and bibliographic reference of this tool must be included in the updated version of the manuscript. According to the NHESS journal policy, software should be clearly referenced to assess the veracity and reproducibility of the work not only by reviewers but also by future readers. Therefore, I request the authors to document the software in the main text, at the end of the manuscript (Section: Code and data availability), and in the References section, ideally including a registered DOI. Alternatively, the authors may provide a temporal review link or an open-source repository (e.g., GitHub, GitLab) in the updated manuscript, with the condition that the software will have a fixed registered version (with a DOI) in the future, before the publication of the journal paper, and will be part of the supplementary material if the manuscript is accepted.

The authors should note that the coding language and platforms used should be open-source. If not, a statement should be provided explaining the current circumstances of the software. This information is crucial for the NHESS Journal to adhere to FAIR (Findable, Accessible, Interoperable, and Reusable) principles."

**1.1.2.** Datasets: Additionally, besides the software, and in order to fulfil the FAIR principles, datasets that are used as inputs in the computations should be also made available to the future readers to reproduce the results. These datasets are at least: the stochastic seismic catalogue, and the diverse exposure models (the baseline one and the counterfactuals).

1.1.2.1. Regarding the stochastic seismic catalogue: the authors are encouraged to state in an updated version of the manuscript if the generation of such a catalogue is a fully original milestone of their study. If so, it must be provided as supplementary material. Or, in the contrary, if it was already generated by previous studies (e.g. Ferrario et al., 2022). Please clarify and cite other existing studies if needed.

1.1.2.2. Regarding the exposure models: please note that I am not suggesting that raw and confidential cadastral datasets are provided as supplementary material. The exposure models that should be provided as supplementary data should be solely expressed in terms of the HAZUS classification scheme and in the required data formats to ensure reproducibility. Moreover, the aggregation area for this exposure model should be, in the best case, the same one reported in the paper.

**1.1.3.** In the submitted version of the paper, it is unclear whether the "models" (i.e., mathematical expressions and workflows) encapsulated in the mentioned tool were originally proposed

by the authors or are existing, conventional approaches that have been reused. The text lacks clarity in this regard. The way Section 2 is written creates the incorrect impression that the conventional mathematical formulations presented therein are part of a completely new method, which, to my understanding, is not accurate. As these approaches already exist and were only slightly modified or adapted in the submitted paper, it is crucial that the authors appropriately cite the studies that initially proposed such approaches. Additionally, there is a notable absence of citations to several state-of-the-art studies in probabilistic seismic risk, such as those conducted by the GEM foundation.

**1.2. Relocation of some text**

**1.2.1.** The Introduction is presented in a quite disorganised manner. The order of presentation of the paragraphs #2, #3 and #4 of the Introduction is counterintuitive. Lines 19-24 mention at a quite early stage the *application* of the proposed approach (authors call it a "model" or "tool") to a particular study area (San Antonio). This idea was already written in the abstract, and is therein (at a very prompt instance) presented without having provided a proper theoretical background beforehand. In the same way, Lines 33- 41 mention for the first time that a *tool* is proposed, but without having provided the motivation or state of the art beforehand. Such a background is actually presented later, in the next paragraph (lines 42-55). Therefore, these lines should be used to describe the state of the art as well as the motivation to then present the need of having developed a "tool" at an easier stage, and lastly, its application to a real case study. Having said that, the suggested ordering would then imply to swap these paragraphs (I use the current numeration) as follows: paragraph #1, #4, #3, #5, #2, #6. Please note the resultant connection between #1 and #4. I do believe that this new ordering will significantly improve the overall quality of the Introduction and will provide a much smother reading experience to the reader.

**1.2.2.** The fact that the content presented in the three sub-sections 2.1, 2.2, 2.3 are located within Section 2 is clearly and once again showing the disorganised structure of the submitted manuscript. The content within these sections should be relocated. Section 2 is intended to comprise generic methodological approaches ("Seismic risk methodology"). Therefore, it results counterintuitive that these 3 sub-sections, which provide quite detailed information, explicitly tailored for the target study area (San Antonio), namely, 2.1 (exposure model), 2.2 (synthetic earthquake catalogue), and 2.3 (generation of ground motion fields) are not part of Section 3 "Case study San Antonio city". Hence, I suggest that the content of these three sections (2.1-2.3), where details about the area of study are given, are correspondingly moved to Section 3.

**1.2.3.** The previous redesign will also require the authors to ensure that Section 2 remains a purely methodological chapter, independent of the study area, as indicated by the selected title, without mixing it with the application to a specific study area. However, the authors should explicitly state what constitutes the true novelty in this "methodology," independent of any study area, or if it is simply an extension of the PBEE to a building portfolio, which is also not novel. Therefore, I kindly suggest that the authors modify the title of Section 2. The new title should either highlight the originality of the method (if applicable) or clarify if it is more of an application of an existing one.

**1.2.4.** Accounting for the former suggestions, Section 3 should be renamed as: *"Counterfactual risk analysis: an application to the city of San Antonio, Chile"* or something similar.

**1.2.5.** Aligned with the former proposed modifications, Figure 9 should be part of the new sub-section *"3.1 Synthetic earthquake catalogue applied to San Antonio"* (or similar). Thus, details related (Lines 304-309) should be also relocated (see next comment). This new proposed order follows the workflow of Fig 1 where the hazard part comes first than the exposure. Thus, this would imply Sect 3.2 to be *"Exposure model"*. This one should comprise two renamed sub-sub-sections: *"3.2.1. Baseline exposure model"*, and *"3.2.2. Counterfactuals in the exposure component"* (or similar). It is very important to point out that the counterfactuals belong only with the exposure and that these assumptions are naturally propagated through the vulnerability analysis.

**1.2.6.** I strongly encourage the authors to follow the same ordering of the workflow proposed in figure 1, when presenting each component. This is not respected in the text. I guess that such an ordering will naturally be expected by any reader. Therefore, I suggest the authors to stick to it to provide a better reading experience than the current one. For instance, it results counterintuitive that the seismic catalogue is the first step mentioned in the workflow, but Figure 9 (that display that step) is the latest one presented and only within Section 4. "Counterfactual risk analysis". Hence, in my opinion, lines 304-309, and Fig. 9 should be relocated to an earlier section (i.e. where the stochastic seismic catalogue and the simulation of the ground motion fields). Please see comment 1.2.5.

**1.2.7.** According to the former suggestions (i.e. comment 1.2.4), section 4 "Counterfactual risk analysis" should be renamed as *"Results"* or something similar.

**1.3. Presentation of figures and tables**

As a general comment, figures and tables should always be self-explanatories with proper captions, legends and headers. Moreover, the figures should be designed for colour-blind people, with stronger contrasts among the selected colours. These simple conditions were not fulfilled in the submitted version of the manuscript.

**1.3.1.** For example, I consider that figure 5, and 6 should be represented following another colour scale. There are several web-sites and apps where the authors can check out for a colour-blind compliance.

**1.3.2.** The captions of the figures are in most of the cases simply too short. As a general rule, all of the figures should have sufficiently self-explanatory captions to be clearly understood standalone. Therefore, I strongly suggest the authors to please revise all of the figures' captions by asking themselves if each individual figure can be understood from the sole graphical and textual information provided. Please, cross-reference some information if needed to avoid being repetitive. For instance, when some figures are made up with acronyms (e.g. Fig 4), and their meanings were already provided somewhere else in the text (e.g. Table 1, and 2), then you should cross-reference Tables 1 and 2 in the Caption of Fig. 4.

A few more hints to assist the authors on how they should complement the figures' captions are provided as follow:

-Figure 1 and 2-b: Please include in the caption the meaning of γ. This explanation was only written in line 224.
-Fig. 2: if you decide to keep this figure, please include the citation of the authors that firstly proposed such a method, including the expression: "after" of "following" or "adapting". This is because these are not original methodological figures, and credit should be given.
-Fig. 3: to which year? According to which data source?
-Fig. 4: Where?
-Fig. 7: Please cross-reference Table 1 and Table 4
-Fig. 8: Please reference FEMA and cross-reference Table 1.
-Fig. 9: Stochastic/ synthetic earthquake catalogue that comprise X events, and that was obtained from which zonation (Poulos, et al., 2019) and by using which sampling strategy?
-Table 1: According to which classification scheme? Where?
-Table 3: According to which classification? Eurocode 8? The DS61? The Turkish norm? Although this might be trivial, this should be indicated in the header.

**1.3.3.** Figure 3-b: there is an apparent mismatching in the proportions assigned to the construction materials for the buildings in San Antonio city. Line 260 reported masonry, wood, steel, and concrete, which account for 41%, 31%, 14%, and 10%, which leads to a sum of 96%. However, the value plotted in Fig. 3-b for "other" does not correspond to the remaining 6%, but ~2%. Where is the other 4%? I suppose this might come from rounding the numbers. Nonetheless, please make sure you correct this, either in the text (line 260) or in the figure.

**1.3.4.** Figure 9-b is not very illustrative. I kindly ask the authors to answer themselves this question: What relevant information would the reader get from this figure besides that the earthquake catalogue for the region is expected to be scatted all over the selected seismic area sources? If the authors' intention is to point out that larger magnitudes are expected preferentially in certain parts, then, a wiser option would be to draw the expected earthquake point sources as circles, whose size depend on the Mw (larger circles represent larger magnitudes), as for instance shown by Rosero-Velásquez et al., (2023) (see Fig. 5) for a similar area of interest.

**2. MAJOR TECHNICAL COMMENTS**

**2.1. Exposure model**

**2.1.1.** The "bottom-up" terminology is not used correctly in the submitted manuscript. The authors should realise that "bottom-up" refers to individual data collection, not to high-resolution exposure models at the block level (that can be also obtained from desktop analysis). Desktop analysis might also constitute top-down approaches when the analysist deals with aggregated data, or by doing assumptions on the number of buildings given the knowledge of dwellings. I kindly suggest the authors to check the study of Pittore et al., (2018) where this differencing is clearly presented. If the authors agree on this clarification, then, such a terminology should be avoided in an updated version of the text.

**2.1.2.** Please include the references (within the text, and the bibliography) of the Chilean Census dataset that was used to determine the population, and the SII database. It is important to have a full citation of this information along with the year of publication of such data. The vintage of these datasets is extremely important for the readers because the risk results that are later presented are only valid for that year. If the years of the two datasets are different, please justify how did you harmonise these discrepancies.

**2.1.3.** The fact that the HAZUS building classes (exclusively designed for both, specific hazard characteristics and building typologies of the United States of America) are adopted in the Chilean context deserves more discussion in the paper. Likewise, the study of Pittore et al., (2018) can also provide some insights about the implications of such an assumption, that at least should be discussed as caveats at the end of the manuscript. Having adopted the HAZUS fragility functions also poses a source of uncertainty in all of the results presented. This should be acknowledged by the authors in the updated version of the manuscript by commenting on the possible mismatch of the intensities adopted by the HAZUS fragility functions to define their limit damage states. These functions were derived from largely contrasting seismic records in comparison with the Chilean one (e.g. Cabrera et al., 2020). Also, please take a look at Hoyos and Hernández, (2021) to briefly comment on this aspect.

**2.1.4.** Please, clearly state how many building classes are used to represent the building stock of San Antonio. Table 1: The authors called it "building typology assignment". However, it is only refereeing to the correspondence between material and 6 building classes. Notably, the authors also say that also the code compliance and height were used to define classes. Therefore, these are not 6 typologies, but materials (I guess of the lateral load resistance system). Be careful with the wording "building typology". These three attributes (disregarding the occupancy and building finishes (these are not mentioned)) describe combinatorics, which, if all them are possible, there would be 72 classes. Is it correct? I suspect that is not the case because I do not think there might be wooden buildings of more than 8 stories. Hence, the authors should specify which combinatorics are not possible to delimitate the exact number of building classes that they propose can model the city.

**2.1.5.** Furthermore, in line 137 it is stated "quality of construction of 4 or 5. This type of parametrization was never described before. How are they assigned? How do 5 quality ranges match 4 design levels? That is not clear nor trivial.

**2.1.6.** How are the "unit construction costs per square meter" defined? I suggest that the authors provide (e.g. in an Appendix or Annex section) the parametrization they follow to classify each of the variables involved to score the costs. Please provide the numerical ranges of numbers of how the 4 variables selected are scored in their approach.

The authors perhaps would like to emphasise a bit more the moderate novelty of such a procedure in obtaining the replacement costs (in comparison with more coarser estimates typically done by GEM). I think that highlighting this part of their approach would show certain improvements in comparison to the SARA exposure model that presented higher limitations (GEM, 2014). However, if that would be one of the "strong" points of the paper,

this aspect should be highlighted along the manuscript (in the introduction, discussion too). If the authors accept this suggestion, please also consider citing the study of Nievas et al., (2022), where the authors also provided certain parametrization to account for risk metrics. Perhaps you would like to discuss the advantages you might have found over these existing studies. I think that by incorporating these aspects, the quality of the paper might increase.

However, the authors should justify better how the replacement costs for commercial and warehouses were obtained. Were the contents accounted for? If so, how? If not, it should be clearly stated that only direct economic replacement costs were addressed, and to comment on that limitation in the discussion section.

**2.1.7.** The fact that "soil type" is included as part of the "exposure component". I do not think this is appropriate. Soil type was only accounted for by the authors in order to stablish a relation with the $Vs_{30}$, which is in turn, a parameter needed to forecast the ground motion fields by the selected GMPE. The former is clearly part of the hazard component, not of the exposure. Ask yourselves: what are the exposed items of interest to earthquakes for which we perform risk analyses? Following this logic, "Soil type" can an "exposed item" of interest to risk only if one can stablishes its own vulnerability by defining certain definitions and thresholds. This is done for instance in other hazards, such as desertification or liquefaction, where soil are the actual exposed items. Hence, one can talk about liquefaction risk and desertification risk of soils. Please note that this definition does not apply for the specific case study of interest, where buildings are the exposed items for which their risk evaluation to earthquakes is the only target. Having said this, I suggest that the soil type and its relations with $Vs_{30}$ are removed from the exposure component, and please include them in sections related to seismic hazard.

**2.1.8.** Include the counterfactual analysis within the exposure section. Please see comment 1.2.5. I strongly suggest that the counterfactual CF4 and CF2 are removed and avoid their further discussion in the results. I really tried to look for the logic behind these assumptions but I could not find it. If there are good reasons, please provide them, because they were not written. One of the alternatives I imagined is the inability of identifying the material of construction and confusing wood with reinforced concreted. Although the mere idea of that is quite unlikely, it might be the case if the input data is not the best or does not provide such an attribute. If that is not the case, and the only idea was to develop an academic exercise, this should be also state, but with logical criteria.

A third option that I can think of the reasoning of CF2 and CF4 is that decision makers are interested in changing the material of construction of existing buildings. Is this because of an ongoing project that looks for demolish existing buildings and build new ones? Besides that, the time variable is not addressed in the analysis, this would require significant extra costs that are not accounted for either. Please provide clarity if that is the case and make the corresponding modifications adding extra costs. If that was neither the case, then, I, once again, support the idea of eliminating these two counterfactuals; and maybe add other ones that are more convincing and towards practical applications.

**2.2. Seismic hazard components**

**2.2.1. Stochastic seismic catalogue:**

2.2.1.1. The authors should be more precise about the manner their stochastic catalogue was generated. As far as I could understood from the text, the epicentral locations of were generated randomly, based on the occurrence rate associated with a predefined seismic zone. This would mean that only independent source parameters are the moment magnitude Mw, longitude X and latitude Y of the epicentre, and other parameters, such as the depth, strike, dip and rake angles, are determined by the geometry derived from (Hayes et al., 2018). However, this slightly differs to other more conventional approaches, such as the offered by the Event-based calculator of the OpenQuake Engine.

2.2.1.2. The authors are missing quite basic terminology in this section. Such as: Poissonian occurrence, Gutternber-Richter relationships; which are underlying assumptions used to generate the catalogue. Also, in line 177 it is said "the site of interest". Is it a fixed unique coordinate? If so, is it a the centroid of the study area?

2.2.1.3. The authors should be aware, a stochastic catalogue is typically generated for a site of interest on the basis of being able to replicate the seismic hazard curve for an explicitly predefined return period. The quantity of the likely seismic events would only come after having performed a sensitivity analysis to ensure that they are representative and stable around the return period of interest. (e.g. Aristizábal et al., 2018). Having said that, I kindly ask the authors to please comment a bit more on the possible implications of having created the catalogue in the manner they decided to.

**2.2.2. GMPE-based ground motion fields**

2.2.2.1. In general section 2.3 contains a lot of unnecessary text-book material that should be reduced as much as possible. General theory is mixed up with the application area (see comment 1.1.2).

2.2.2.2. Complementarily, I suggest that the authors briefly comment on the logic behind having adopted a single GMPE. Perhaps the studies of Hussain et al., (2020) and Gómez Zapata et al., (2022) might help to justify that selection for the study area of interest. This limitation should be retaken in the Discussion section at the end of the manuscript.

2.2.2.3. I really like that the authors are using a local correlation model. This should be highlighted as one of the novelties of the approach. In order to strength out this aspect, I suggest that the authors select one of the scenarios for a given return period of interest (e.g. Rosero-Velásquez et al., 2023; Indirli et al., 2011) and present a figure of how a single realisation of the spatially correlated ground motion field coming from the respective earthquake rupture would look like.

**2.2.3. Seismic microzonation/ $Vs_{30}$ estimates**

2.2.3.1. Was the microzonation reported in Mendoza et al., (2018) used in the SIGAS project by Sernageomin? If so, I suggest to also cite that paper or more recent one to allow the readers investigate the techniques and assumptions involved in those estimations. Moreover, please indicate if the area where the soil profiles were inferred fully covers the entire study area. If not, then, what source of information was used to complete the remaining parts? USGS datasets?

2.2.3.2. The authors state that the soil type was assigned to each census block according to its location, and then, a single $Vs_{30}$ was assigned too. Because of that statement, I wonder if an irregular grid of sites was used to estimate the ground motion values (using a GMPE). I pose that assumption because a node within an irregular grid will be formed from every block's centroid. If that is not the case, please rephrase the former sentences, and if a regular grid was employed instead, please indicate what is the spacing intervals between sites. Please comment on this in the discussion if the separation between sites might be too coarse, and how to improve that in the future, or if in the contrary, it is sufficient for the purpose of the study.

**2.3. Damage and loss assessment.**

Section "2.4 Damage analysis" contains quite a lot of text-book material that should be reduced as much as possible. I suggest to rename this subsection "physical vulnerability assessment". This way, the content of 2.5 could be merged with 2.4 into a single more concise one. In this section, please provide clarity of the loss ratios assumed and if they are building dependent or independent. If that is too large, it can be allocated in an annex.

**2.4. Results, discussion and conclusions**

**2.4.1.** As stated before, I strongly suggest the creation of a discussion section where the limitations of some of the assumptions are clearly acknowledged.

**2.4.2.** Results and conclusions are nor surprising. The authors should find a way to deeply change the manner they are provided in a more convincing manner while highlighting the possible novelties of their approach and applications.

**2.4.3.** I suggest the creation of at least a new figure where the risk estimates (physical damage, casualties, and direct economic ones) are geographically shown. Having a spatial representation will be useful to the reader. For this, the authors could consider to use a single (or several) ground motion realization from a return period of interest (see comment 2.2.2.3) to do so. Percentiles representations would be also useful (e.g. Goda et al., 2021). This is just a simple recommendation to have a comparative baseline between the exposure models and risk assessment.

**2.5. Suggested publications to be cited in the revised version**

I kindly suggest to at least include the following references: (Indirli et al., 2011; Gómez Zapata et al., 2022; Aldea et al., 2022; Rosero-Velásquez et al., 2022; Baquedano-Juliá et al., 2023; Geiß et al., 2023; Rosero-Velásquez et al., 2023). They constitue relevant background information that should be presented along the description of the study area (hopefully in the newly updated Chapter 3 (see comment 1.2.2 of this communication).

**3. EDITORIAL COMMENTS:**

**3.1. Suggestion to rephrase some sentences.**

**3.1.1.** I suggest to please be more specific in the way the titles and subtitles are written. For instance, "3.1. Baseline" (of what?) "2.3. Seismic analysis" (for what?) Do you mean the generation of GMPE-based ground motion fields?

Please, check out all the titles and rewrite them to turn them into self-explanatory sentences and as specific as possible. Please, see comment 1.2.2.

**3.2. English quality.**

There are several typos that might be solved by the Editorial office of the journal if it is accepted for publication later on. There are several issues that must be fixed. Hereby, I just mention a few as examples. Still, I strongly recommend this paper goes through an extensive English edition and proofreading (grammar and punctuation).

**3.2.1.** With the next file upload request, please update the copyright statement on the figures' captions to: © OpenStreetMap contributors YEAR. Distributed under the Open Data Commons Open Database License (ODbL) v1.0.

**3.2.2.** Line 19: I suggest you modify the word "model" by "workflow" or similar. My understanding is that the authors do not propose a "new model", as the different components or modules already exist and are widely used in research and insurance.

**3.2.3.** Line 23: Please change the occupancy classes "commerce" and "industry" to "commercial" and "industrial". This is a common nomenclature in the GEM v 2.0 and 3.0 taxonomies. Moreover, I advise the authors to avoid using "etc" therein.

**3.2.4.** Line 22: I suggest you use a "connector expression" right before "we generate", such as: "For this, we generate". This will provide the reader a smother experience.

**3.2.5.** Line 25: change "the entire" catalog to "an entire" or similar. This is because are many ways to construct several seismic catalogues for the same study area.

**3.2.6.** Line 49: please be specific: cross-correlation models between the residuals of the forecasted intensity measures from ground motion prediction equations.

**3.2.7.** Line 53: change "However" to other similar word. The same word was written in line 49.

**3.2.8.** If the authors decide to keep the header "2.5. Consequences analysis", please change to "Consequence analyses". This is because the word "consequence" has an adjective attribute herein and therefore should be singular. Moreover, the plural or "analysis" is "analyses".

**3.2.9.** Line 60: The reference Villar-Vega et al., (2017) is not appropriate for the sentence written. That study provides seismic fragility functions, and has nothing to do with spatial proxies. Please delete it from it. Do the authors perhaps refer to a similar reference (Yepes-Estrada et al., 2017) that was also published in the framework of the SARA project?

**3.2.10.** Line 97: the correct form here would be "where", not "were". However, "for which" fits better. On this same line, I kindly suggest to replace "people" by "inhabitants" or "residents". The word "people" is too general.

**3.2.11.** Line 140 and similar: I suggest you use "built-up area" instead of "built area".

**3.2.12.** Header 3: Please include ":" after "study".

**3.2.13.** Lines 170 & 175: please avoid redundancy. Consider keeping the idea once.

**3.2.14.** Lines 243 & 253: please avoid redundancy. Consider keeping the idea once.

**References**

Aldea, S., Heresi, P., and Pastén, C.: Within-event spatial correlation of peak ground acceleration and spectral pseudo-acceleration ordinates in the Chilean subduction zone, Earthquake Engineering & Structural Dynamics, 51, 2575–2590, https://doi.org/10.1002/eqe.3674, 2022.

Aristizábal, C., Bard, P.-Y., Beauval, C., and Gómez, J. C.: Integration of Site Effects into Probabilistic Seismic Hazard Assessment (PSHA): A Comparison between Two Fully Probabilistic Methods on the Euroseistest Site, Geosciences, 8, https://doi.org/10.3390/geosciences8080285, 2018.

Baquedano-Juliá, P., Ferreira, T. M., Arriagada-Luco, C., Sandoval, C., Palazzi, N. C., and Oliveira, D. V.: Multi-vulnerability analysis for seismic risk management in historic city centres: an application to the historic city centre of La Serena, Chile, Natural Hazards, https://doi.org/10.1007/s11069-023-06008-8, 2023.

Cabrera, T., Hube, M., and Santa María, H.: Empirical fragility curves for reinforced concrete and timber houses, using different intensity measures, 17thWorld Conference on Earthquake Engineering, 17WCEE, Sendai, Japan, 2020.

Ferrario, E., Poulos, A., Castro, S., de la Llera, J. C., and Lorca, A.: Predictive capacity of topological measures in evaluating seismic risk and resilience of electric power networks, Reliability Engineering & System Safety, 217, 108040, https://doi.org/10.1016/j.ress.2021.108040, 2022.

Geiß, C., Priesmeier, P., Aravena Pelizari, P., Soto Calderon, A. R., Schoepfer, E., Riedlinger, T., Villar Vega, M., Santa María, H., Gómez Zapata, J. C., Pittore, M., So, E., Fekete, A., and Taubenböck, H.: Benefits of global earth observation missions for disaggregation of exposure data and earthquake loss modeling: evidence from Santiago de Chile, Natural Hazards, 119, 779–804, https://doi.org/10.1007/s11069-022-05672-6, 2023.

GEM: Report on the SARA Exposure and Vulnerability Workshop in Medellin, Colombia, 2014.

Goda, K., Risi, R. D., Luca, F. D., Muhammad, A., Yasuda, T., and Mori, N.: Multi-hazard earthquake-tsunami loss estimation of Kuroshio Town, Kochi Prefecture, Japan considering the Nankai-Tonankai megathrust rupture scenarios, International Journal of Disaster Risk Reduction, 54, 102050, https://doi.org/10.1016/j.ijdrr.2021.102050, 2021.

Gómez Zapata, J. C., Zafrir, R., Pittore, M., and Merino, Y.: Towards a Sensitivity Analysis in Seismic Risk with Probabilistic Building Exposure Models: An Application in Valparaiso, Chile Using Ancillary Open-Source Data and Parametric Ground Motions, ISPRS International Journal of Geo-Information, 11, https://doi.org/10.3390/ijgi11020113, 2022.

Hayes, G. P., Moore, G. L., Portner, D. E., Hearne, M., Flamme, H., Furtney, M., and Smoczyk, G. M.: Slab2, a comprehensive subduction zone geometry model, Science, 362, 58–61, https://doi.org/10.1126/science.aat4723, 2018.

Hoyos, M. C. and Hernández, A. F.: Impact of vulnerability assumptions and input parameters in urban seismic risk assessment, Bulletin of Earthquake Engineering, 19, 4407–4434, https://doi.org/10.1007/s10518-021-01140-x, 2021.

Hussain, E., Elliott, J. R., Silva, V., Vilar-Vega, M., and Kane, D.: Contrasting seismic risk for Santiago, Chile, from near-field and distant earthquake sources, Natural Hazards and Earth System Sciences, 20, 1533–1555, https://doi.org/10.5194/nhess-20-1533-2020, 2020.

Indirli, M., Razafindrakoto, H., Romanelli, F., Puglisi, C., Lanzoni, L., Milani, E., Munari, M., and Apablaza, S.: Hazard Evaluation in Valparaíso: the MAR VASTO Project, Pure and Applied Geophysics, 168, 543–582, https://doi.org/10.1007/s00024-010-0164-3, 2011.

Mendoza, L., Ayala, F., Fuentes, B., Soto, V., Sáez, E., Yañez, G., Montalva, Gonzalo., Gález, C., Sepúlveda, N., Lazo, I., and Ruiz, J.: Estimación cuantitativa de la amenaza sísmica en base a métodos geofísicos: aplicación a las localidades costeras del segmento los Vilos – San Antonio, 50 Congreso SOCHIGE., Valparaiso, Chile, 2018.

Nievas, C. I., Pilz, M., Prehn, K., Schorlemmer, D., Weatherill, G., and Cotton, F.: Calculating earthquake damage building by building: the case of the city of Cologne, Germany, Bulletin of Earthquake Engineering, https://doi.org/10.1007/s10518-021-01303-w, 2022.

Pittore, M., Haas, M., and Megalooikonomou, K. G.: Risk-Oriented, Bottom-Up Modeling of Building Portfolios With Faceted Taxonomies, Frontiers in Built Environment, 4, 41, https://doi.org/10.3389/fbuil.2018.00041, 2018.

Rosero-Velásquez, H., Gómez Zapata, J. C., and Straub, D.: Comparative assessment of models of cascading failures in power networks under seismic hazard, in: Proceedings of the 32nd European Safety and Reliability Conference, ESREL2022, Dublin, Ireland, 2022.

Rosero-Velásquez, H., Monsalve, M., Gómez Zapata, J. C., Ferrario, E., Poulos, A., de la Llera, J. C., and Straub, D.: Risk-informed representative earthquake scenarios for Valparaíso and Viña del Mar, Chile, Natural Hazards and Earth System Sciences Discussions, 2023, 1–25, https://doi.org/10.5194/nhess-2023-186, 2023.

Villar-Vega, M., Silva, V., Crowley, H., Yepes, C., Tarque, N., Acevedo, A. B., Hube, M. A., Gustavo, C. D., and María, H. S.: Development of a Fragility Model for the Residential Building Stock in South America, Earthquake Spectra, 33, 581–604, https://doi.org/10.1193/010716EQS005M, 2017.

Yepes-Estrada, C., Silva, V., Valcárcel, J., Acevedo, A. B., Tarque, N., Hube, M. A., Coronel, G., and María, H. S.: Modeling the Residential Building Inventory in South America for Seismic Risk Assessment, Earthquake Spectra, 33, 299–322, https://doi.org/10.1193/101915EQS155DP, 2017.

---

## Author Comment (AC1)

**Response to reviewer RC1:**

We thank the reviewer for the suggestions to improve the document. At the same time, we regret that the exposition was not clear enough for the reviewer to fully understand the key aspect of the document's contribution and novelty.

What is novel about our approach is the combined application of a fully probabilistic risk assessment methodology at a city level, along with a very high resolution in the exposure model. By fully probabilistic we mean that our main objective is to obtain the final, unconditional, probability distribution over all potential seismic consequences (we study physical damage, economic losses, and casualties), and not the distribution conditioned to a particular seismic scenario or event, as is commonly found in the literature. On the one hand, we calculate consequences event by event considering thousands of seismic scenarios, taking care of the spatial correlation of the seismic intensity measures in each scenario. On the other hand, models at a city level are scarce, the more so for finer resolutions. Our resolution is the finest, as we consider each individual building unit in the city with its particular characteristics. Additionally, our counterfactual scenario analysis provides valuable insights into the potential impact of changes in building classes on the distribution of annual losses for various consequence variables. The comprehensive information provided by this fully probabilistic approach, i.e. annual distribution of physical damage, economic losses and/or casualties, is significantly useful for decision makers, serving as a substantial complement to the risk metrics derived from scenario-based evaluations. In fact, as acknowledge in this manuscript, part of this study is required and funded by the Chilean National Science Foundation (ANID) to work closely with the Ministry of Housing (MINVU) to provide these findings as tools to orient decision making at the public policy level.

While extensive research can be found regarding risk assessment in urban areas around the world (De Risi et al., 2019; Basaglia et al., 2018) and also South America (Villar-Vega and Silva, 2017; Feliciano et al., 2023) and Chile (Hussain et al. 2020; Baquedano et al., 2023; Geiß et al., 2023; Gómez et al., 2023 ), which include significant advancements in the exposure models, and fragility, most of them share a common denominator: **they are conditioned to a specific seismic event**. For example, the first two studies recommended by the reviewer (https://doi.org/10.1029/2021EF002388, https://doi.org/10.1177/87552930221134950,) emphasize high-resolution exposure models, but within the constraints of a particular seismic scenario. Similarly, another study (https://doi.org/10.3389/feart.2020.575048) provides counterfactual analysis but is also conditioned to a specific scenario. In contrast, risk studies considering the temporal distribution of seismic events are limited, mainly because of the high computational cost involved related to the sampling of thousands of earthquake rupture scenarios and subsequent ground motion scenarios. Although there are some examples globally and within South America, including Chile (Yepes-Estrada and Silva, 2017; Petersen et al., 2018), there's a significant gap in progressing risk assessment methodologies to encompass multiple earthquake scenarios.

While recognizing the value of risk assessment conditioned to specific scenarios, especially in risk communication, we emphasize the benefits of a fully probabilistic approach, i.e., considering multiple seismic events and calculating consequences event-by-event.

First, by assessing consequences of a seismic scenario with a specific average return period, and then aggregating losses through classical PSHA, does not take into consideration the spatial correlation of the intra-event residuals (Jayaram and Baker 2009), nor the correlation of loss ratio

between buildings of the same vulnerability class, even if the intensity measures were initially spatially and spectrally correlated.

Second, the consequences of an earthquake scenario that has an average return period of $X$ years are not the same as the consequences that have an average return period of $X$ years (Ellingwood, 2009). Furthermore, using Poisson's assumption, those consequences that have a return period of $X$ years are expected to be exceeded on average at least once, which is not the same as the probability of having that consequence only once.

Third, to consider both aspects appropriately, a fully probabilistic event-based approach is needed, where losses are calculated event by event, which leads to a loss exceedance curve representative of the whole exposure model (Silva et al., 2015). This approach leads to higher probabilities of exceeding large losses (e.g., Park et al., 2007; Silva et al., 2014). Additionally, this effect is also present in hazard studies. When considering multiple seismic scenarios, seismic hazard estimates tend to be higher, because in "earlier studies the ground-motion variability was either completely neglected or treated in a way that artificially reduced its influence on the hazard" (Bommer & Abrahamson, 2006). Bommer & Crowley (2006) also emphasizes that to capture the variability in ground motion, the recommended approach involves modeling a "large number of earthquake scenarios that sample the magnitude and spatial distributions of the seismicity, and also the distribution of ground motions for each event".

We use Monte Carlo simulations to generate a stochastic synthetic earthquake catalog, where thousands of earthquakes magnitudes are sampled from a truncated exponential distribution with parameters based on the most updated Chilean recurrence model (i.e. Guttenberg-Richter law). Also, each earthquake magnitude is spatially sampled from a uniform distribution in the subduction interface defining a hypocenter for each sample, and therefore defining a rupture surface. Next, for each rupture of the earthquake catalog, the ground motion model is sampled to obtain spatially distributed intensity measures for each scenario, and therefore its consequences. Finally, using the total probability theorem, all the scenarios are integrated, enabling us to compute annual consequence distributions that reflect the city's seismicity (Crowley and Bommer, 2006; Baker and Cornell, 2008; Jayaram and Baker, 2010; Allen et al., 2022; Ferrario et al., 2022).

In addition to employing a fully probabilistic approach, we developed a highly detailed exposure model. The actual exposed building stock of the city is constructed starting from each building unit within a city block, utilizing the most detailed publicly available information. For each building unit within a city block, we assign a building class based on available data such as material, use, year of construction, etc. Consequently, we estimate the consequences event by event, taking into account the specific seismic intensity measure for each census block in every seismic scenario, as well as per square meter built for each building class within that block.

Our counterfactual scenario analysis is just one example of what models of this sort can deliver. The scenarios discussed in this research are meant to reflect shifts not in the city's expansion but in its densification or changes in land use. This provides valuable insights regarding the potential impact that such changes may generate in terms of the annual losses' distribution calculated for the different consequence variables between the current exposure and the counterfactuals. The results provided by the counterfactuals are not trivial, primarily because they depend on the current exposure and hazard, which is a real city case, and second, because the result is not the same across different consequence variables. On the one hand, the impact of changing the predominant building class (which represents 27% of the total exposed built area), has significant

but different impact on physical damage, casualties and economic losses (48%, 80% and 17-24%). However, changing the second predominant building class (which represents 21% of the exposed built area) has an almost negligible impact on the analyzed consequence variables. These results are not trivial and can significantly impact the decisions on promoting new constructions in areas currently dominated by one building class or another.

In summary, our research aims to address a significant gap in current seismic risk assessment studies. We adopt a comprehensive probabilistic approach by considering thousands of seismic scenarios and calculating losses event-by-event. Our focus is on assessing probability distributions for various consequence variables within specific time frames in urban environments, including different counterfactual scenarios. This effort enhances our understanding and contributes to advancing seismic risk assessment methodologies, providing a valuable complement to scenario-based studies.

Although we are not providing a particular new methodology for seismic risk assessment, **we are integrating the different, state-of-the-art pieces involved in a fully probabilistic risk assessment,** at a city level, along with a very high resolution in the exposure model. We provide unique results on expected consequences (i.e. physical damage, economic losses and casualties) related to seismic risk in the San Antonio study area. We firmly believe that decision makers such as city planners, construction sector regulators, and insurance regulators, urgently require information on annual distributions of an array of disaster consequence variables (Smith, 2004). Many decisions require a full risk assessment, i.e., consideration of the whole probability distribution for any timeframe. This is the case for example of the insurance industry, where specific percentiles of the loss distribution or the mean (expected losses) are considered relevant risk measures, among others (Goda et al., 2015; Yoshikawa and Goda, 2014). Moreover, with the complete consequence distribution at ones' disposal, one can derive numerous risk metrics, including average return periods, probability of exceedance within a time frame, etc.

Regarding the reviewer's specific comments, they could be effectively addressed in a revised manuscript, as they mainly involve clarifying points rather than indicating methodological or technical deficiencies. Furthermore, we believe that in a revised version, we could clarify the aspects that concerned the reviewer regarding novelty and contribution, which were not sufficiently clear in the previous version. Should the editor invite us to submit a revised version of the manuscript, we would gladly provide a detailed response to the reviewer's specific comments.

Allen, E., Chamorro, A., Poulos, A., Castro, S., de la Llera, J. C., & Echaveguren, T. (2022). Sensitivity analysis and uncertainty quantification of a seismic risk model for road networks. *Computer-Aided Civil and Infrastructure Engineering*, *37*(4), 516-530.

Baker, J. W., & Cornell, C. A. (2008). Uncertainty propagation in probabilistic seismic loss estimation. *Structural Safety*, *30*(3), 236-252.

Baquedano-Juliá, P., Ferreira, T. M., Arriagada-Luco, C., Sandoval, C., Palazzi, N. C., & Oliveira, D. V. (2023). Multi-vulnerability analysis for seismic risk management in historic city centres: an application to the historic city centre of La Serena, Chile. *Natural Hazards*, 1-44.

Basaglia, A., Aprile, A., Spacone, E., & Pilla, F. (2018). Performance-based seismic risk assessment of urban systems. *International Journal of Architectural Heritage*, 12(7-8), 1131-1149.

Bommer, J. J., & Abrahamson, N. A. (2006). Why do modern probabilistic seismic-hazard analyses often lead to increased hazard estimates?. *Bulletin of the Seismological Society of America*, *96*(6), 1967-1977.

Bommer, J. J., & Crowley, H. (2006). The influence of ground-motion variability in earthquake loss modelling. *Bulletin of earthquake engineering*, *4*, 231-248.

Crowley, H., & Bommer, J. J. (2006). Modelling seismic hazard in earthquake loss models with spatially distributed exposure. *Bulletin of Earthquake Engineering*, *4*, 249-273.

De Risi, R., Penna, A., & Simonelli, A. L. (2019). Seismic risk at urban scale: the role of site response analysis. *Soil Dynamics and Earthquake Engineering*, *123*, 320-336.

Ellingwood, B. R. (2009). Assessment and mitigation of risk from low-probability, high-consequence hazards. *Australian Journal of Structural Engineering*, *9*(1), 1-7.

Feliciano, D., Arroyo, O., Cabrera, T., Contreras, D., Valcárcel Torres, J. A., & Gómez Zapata, J. C. (2023). Seismic risk scenarios for the residential buildings in the Sabana Centro province in Colombia. *Natural Hazards and Earth System Sciences*, 23(5), 1863-1890.

Ferrario, E., Poulos, A., Castro, S., de la Llera, J. C., & Lorca, A. (2022). Predictive capacity of topological measures in evaluating seismic risk and resilience of electric power networks. *Reliability Engineering & System Safety*, *217*, 108040.

Geiß, C., Priesmeier, P., Aravena Pelizari, P., Soto Calderon, A. R., Schoepfer, E., Riedlinger, T., ... & Taubenböck, H. (2023). Benefits of global earth observation missions for disaggregation of exposure data and earthquake loss modeling: evidence from Santiago de Chile. *Natural Hazards*, 119(2), 779-804.

Goda, K., Wenzel, F., Daniell, J., Beer, M., Kougioumtzoglou, I. A., & Patelli, E. (2015). Insurance and reinsurance models for earthquake. *Encyclopedia of earthquake engineering*, 1184-1206.

Gómez Zapata, J. C., Zafrir, R., Pittore, M., & Merino, Y. (2022). Towards a Sensitivity Analysis in Seismic Risk with Probabilistic Building Exposure Models: An Application in Valparaíso, Chile Using Ancillary Open-Source Data and Parametric Ground Motions. *ISPRS International Journal of Geo-Information*, 11(2), 113.

Hussain, E., Elliott, J. R., Silva, V., Vilar-Vega, M., & Kane, D. (2020). Contrasting seismic risk for Santiago, Chile, from near-field and distant earthquake sources. *Natural Hazards and Earth System Sciences*, 20(5), 1533-1555.

Jayaram, N., & Baker, J. W. (2009). Correlation model for spatially distributed ground-motion intensities. *Earthquake Engineering & Structural Dynamics*, *38*(15), 1687-1708.

Jayaram, N., & Baker, J. W. (2010). Efficient sampling and data reduction techniques for probabilistic seismic lifeline risk assessment. *Earthquake Engineering & Structural Dynamics*, *39*(10), 1109-1131.

Park, J., Bazzurro, P., & Baker, J. W. (2007). Modeling spatial correlation of ground motion intensity measures for regional seismic hazard and portfolio loss estimation. *Applications of statistics and probability in civil engineering*, *2*, 1-8.

Petersen, M. D., Harmsen, S. C., Jaiswal, K. S., Rukstales, K. S., Luco, N., Haller, K. M., ... & Shumway, A. M. (2018). Seismic hazard, risk, and design for South America. *Bulletin of the Seismological Society of America*, 108(2), 781-800.

Silva, V., Crowley, H., Pagani, M., Monelli, D., & Pinho, R. (2014). Development of the OpenQuake engine, the Global Earthquake Model's open-source software for seismic risk assessment. *Natural Hazards*, *72*, 1409-1427.

Silva, V., Crowley, H., Varum, H., & Pinho, R. (2015). Seismic risk assessment for mainland Portugal. *Bulletin of Earthquake Engineering*, *13*, 429-457.

Smith, W. (2004). The decision support model for risk management: a conceptual approach. *Bulletin of the New Zealand Society for Earthquake Engineering*, *37*(4), 149-155.

Villar-Vega, M., & Silva, V. (2017). Assessment of earthquake damage considering the characteristics of past events in South America. *Soil Dynamics and Earthquake Engineering*, 99, 86-96.

Yepes-Estrada, C., & Silva, V. (2017, January). Probabilistic seismic risk assessment of the residential building stock in South America. *In 16th World Conference on Earthquake Engineering*.

Yoshikawa, H., & Goda, K. (2014). Financial seismic risk analysis of building portfolios. *Natural Hazards Review*, *15*(2), 112-120.

---

## Author Comment (AC2)

**Response to reviewer RC2:**

We are grateful for the extensive comments and suggestions provided by the reviewer, as some of them have the potential to enhance the quality of the document. However, most of the suggested comments would be more suitable for an article on seismic risk conditioned to an event, and not necessarily for this case which follows a fully probabilistic approach. Additionally, we acknowledge with some disappointment that the reviewer encountered challenges in recognizing the novelty of the document.

What is novel about our approach is the combined application of a fully probabilistic risk assessment methodology at a city level, along with a very high resolution in the exposure model. By fully probabilistic we mean that we are after the final, unconditional, probability distribution over seismic consequences (we study physical damage, economic losses, and casualties), and not the distribution conditional on a particular seismic scenario or event, as is commonly found in the literature. On the one hand, we calculate consequences event by event considering thousands of seismic scenarios, taking care of the spatial correlation of the seismic intensity measures in each scenario. On the other hand, models at a city level are scarce, the more so the finer the resolution. Our resolution is the finest, as we consider each individual building in the city with its particular characteristics. Additionally, our counterfactual scenario analysis provides valuable insights into the potential impact of changes in building classes on the distribution of annual losses for various consequence variables. The comprehensive information provided by this fully probabilistic approach, i.e. annual distribution of physical damage, economic losses and/or casualties, is significantly useful for decision makers, serving as a substantial complement to the risk metrics derived from scenario-based evaluations. (Please refer to response to reviewer 1 for more details).

Regarding software and data availability (comment 1.1), the tool referred to in the document corresponds to a code developed by the researchers to address the questions posed in this article, and it is not intended to be used or patented as software. Both, reviewers and the general public, could develop their own code to replicate the results of this study, considering the flowchart in Figure 1, the available models referenced in the article and the public datasets available, making this manuscript completely reproducible. The data to generate the exposure model is available upon request, but we believe it is not a contribution that adds to the results of this article. In the case of the synthetic earthquake catalog, it is stochastic, i.e., corresponds to thousands of samples of earthquakes magnitudes and locations, thus the dataset is not a contribution by itself. However, in a revised version of the manuscript a more detailed description will be included.

Most of the comments provided by the reviewer could be resolved in a revised version of the manuscript, as they do not pertain to methodological or technical deficiencies but rather to clarifying aspects that were not sufficiently clear to the reader in the current version. However, it is important to note that the approach followed in this article is fully probabilistic, and most of the comments suggested by the reviewer align more closely to an event-based risk assessment, so not all of the comments are applicable for this article. Should the editor invite us to submit a revised version of the manuscript, we would gladly provide a detailed response to the reviewer's specific comments.